# Estimating the wake deflection downstream of a wind turbine in different atmospheric stabilities: An LES study

Lukas Vollmer, Gerald Steinfeld, Detlev Heinemann, and Martin Kühn

ForWind, Carl von Ossietzky Universität Oldenburg, Institute of Physics, Ammerländer Heerstr. 136, 26129 Oldenburg, Germany.

*Correspondence to:* Lukas Vollmer (lukas.vollmer@uni-oldenburg.de)

**Abstract.** An intentional yaw misalignment of wind turbines is currently discussed as one possibility to increase the overall energy yield of wind farms. The idea behind this control is to decrease wake losses of downstream turbines by altering the wake trajectory of the controlled upwind turbines. For an application of such an operational control, precise knowledge about the inflow wind conditions, the magnitude of wake deflection by a yawed turbine and the propagation of the wake is crucial. The dependency of the wake deflection on the ambient wind conditions as well as the uncertainty of its trajectory are not sufficiently covered in current wind farm control models. In this study we analyze multiple sources that contribute to the uncertainty of the estimation of the wake deflection downstream of yawed wind turbines in different ambient wind conditions. We find that the wake shapes and the magnitude of deflection differ in the three evaluated atmospheric boundary layers of neutral, stable and unstable thermal stability. Uncertainty in the wake deflection estimation increases for smaller temporal averaging intervals. We also consider the choice of the method to define the wake center as a source of uncertainty as it modifies the result. The variance of the wake deflection estimation increases with decreasing atmospheric stability. Control of the wake position in a highly convective environment is therefore not recommended.

## 1 Introduction

The performance of a wind farm does not only depend on the ability of its wind turbines to convert available kinetic energy into electric energy but is also largely influenced by the fluctuation of the atmospheric winds and the wakes created by the turbines. Wind turbine wakes are areas of lower wind speed and enhanced turbulence that result from the extraction of kinetic energy from the flow by the turbine and can have a significant impact on the wind conditions up to 10-15 rotor diameters downstream. To minimize the losses due to wind turbine wakes, the wind rose measured at a location is usually taken into account during the design process of the wind farm layout. However, in most locations, in particular in mid-latitudes with alternating low and high pressure systems, the unsteady wind direction creates a high occurrence of situations for which wake losses remain large.

Multiple studies, e.g. Barthelmie and Jensen (2010); Hansen et al. (2012), have shown that the wake losses in wind farms depend on the turbulence intensity of the ambient wind, with decreasing efficiency of the wind farm for low turbulence. Sources of turbulence in the atmospheric boundary layer are mechanical shear and buoyancy. The latter depends mainly on the

thermal stratification and can also be a sink of turbulence. In a stably stratified atmospheric boundary layer (SBL) turbulence is suppressed by the stable thermal stratification that decelerates the vertical movement of air masses while in a convective atmospheric boundary layer (CBL) the source of energy at the bottom of the atmosphere enhances the turbulent motion. Studies of atmospheric stability have shown that convective and stable conditions occur at least as often as neutral conditions (NBL) at on-
shore (Vanderwende and Lundquist, 2012; Wharton and Lundquist, 2012) and offshore (Barthelmie and Jensen, 2010; Hansen et al., 2012; Dörenkämper et al., 2014) wind farms and that wind farms are least efficient in stable conditions (Barthelmie and Jensen, 2010; Hansen et al., 2012; Dörenkämper et al., 2014).

The observation of a change of wind farm performance with different atmospheric stability has been supported by wind tunnel
experiments and numerical studies. It has been either related to a generally different level of turbulence (Hancock and Zhang, 2015) or to the presence of large scale fluctuations that enhance the so-called meandering of the wakes in less stable situations (Machefaux et al., 2015a; Larsen et al., 2015; Keck et al., 2014; España et al., 2011). Emeis (2010) and Abkar and Porté-Agel (2013) argue that the thermal stratification above the wind farm becomes important for large wind farms as the vertical momentum transport becomes the only kinetic energy source to refill the wake deficit. Apart from the energy yield, the structural
loads on turbines in the wake also differ with atmospheric stability as they are influenced by up- and downdrafts and large coherent structures in a CBL (Churchfield et al., 2012) and by sharp velocity gradients in an SBL (Bromm et al., 2016).

With increasing capacity of wind turbines the value of every additional percentage of energy that can be harvested from the wind becomes larger. As a consequence the interest to increase the power output for unfavorable wake situations is growing.
Recent studies focus on the control of upwind turbines to minimize wake losses of downwind turbines by either reducing the induction (Corten and Schaak, 2003) or by an intentional yaw angle of the turbine to the wind direction (Medici and Dahlberg, 2003; Jimenez et al., 2010; Fleming et al., 2014). The first approach aims on less extraction of energy from the wind by the upwind turbine and therefore more remaining energy that can be extracted by downwind turbines. The second approach relies on an induction of a cross stream momentum by the upwind turbine to change the trajectory of the wake with the goal to deflect
it away from the downwind turbine. While in both approaches the upwind turbine experiences a loss in power and possibly an increase in structural loads, the additional gain at the downwind turbine is assumed to exceed this loss, thus leading to a surplus of total power output of the wind farm. Based on this assumption, simple models for a joint control of wind turbines to increase power output during operation for a fixed layout have been proposed (Annoni et al., 2015; Gebraad et al., 2016). Fleming et al. (2016) even suggest including power yield optimization by wind farm control in the design process of new wind
farm layouts.

Crucial for wind farm control models is a proper description of the wake trajectory as a wrong description would almost certainly lead to a reduction of energy yield of the wind farm due to the lower energy yield of the upwind turbines. However, magnitudes of the wake deflection differ already in the parameterizations of Jimenez et al. (2010) and Gebraad et al. (2016).
Possible reasons for the differences include the use of different turbine models, the method to extract the wake trajectory from

the measured wind field and the ambient wind conditions. Apart from the differences in the description of the mean wake trajectory, an aspect that is not considered yet in current wind farm control models is the stochastic nature of the wake trajectory. Keck et al. (2014) show not only that the movement of the wake becomes more and more stochastic for small averaging intervals, but also that these motions are linked to atmospheric stability. Considering that the potential to improve wind farm

efficiency through wind farm control appears to be dependent on atmospheric stability, little knowlegde exists on how the control would need to adapt to changes of the wind conditions as influenced by atmospheric stability.

In this study we analyze multiple sources that contribute to the uncertainty of the estimation of the wake deflection downstream of yawed wind turbines in different ambient wind conditions. The ambient wind conditions are created by Large Eddy

Simulations (LES) of atmospheric boundary layers of neutral, stable and unstable stability. The simulations are run with the same mean wind speed and wind direction but changing the stability produces differences in the shear and turbulence of the wind. The wind turbine wakes are created by enhanced actuator disc models with rotation (Dörenkämper et al., 2015b). We use the data from these simulations not only to analyze if the stability changes the magnitude of the wake deflection but also to compare different fitting routines to extract the wake center. In addition to these aspects, that we already consider as contribu-

tors to the uncertainty of the wake deflection estimation, we also look at the influence of different temporal averaging intervals on our results.

## 2  Methods

### 2.1  Estimating the wake deflection

We assume that the wake position $\mu_y$ at a certain distance downstream of a wind turbine can be predicted when the hub height

wind direction $\alpha_h$ and the wake deflection $\Delta y_\gamma$ are known.

$$\mu_y = y_0(\alpha_h) + \Delta y_\gamma \tag{1}$$

where $y_0$ is the displacement of the wake in a fixed coordinate system by the change of wind direction (Fig. 1).

The advantage of LES is that the wake position and the wind direction can be assessed directly from the flow field to esti-

mate the unknown deflection of the wake by the yawed turbine. For a fixed thrust coefficient, turbine site, wind speed and wind direction, the wake deflection is assumed to be a function of the yaw angle $\gamma$ and the atmospheric stability, e.g. given by the Monin-Obhukov length $L$.

$$\Delta y_\gamma = \Delta y_\gamma(\gamma, L) \tag{2}$$

The relationship of $\Delta y_\gamma$ on the yaw angle and the atmospheric stability is estimated from multiple LES with different $\gamma$ and $L$.

$$<\Delta y_\gamma>|_{\gamma,L} = <\mu_y(f_i)> - <y_0(\alpha_h)> \tag{3}$$

Here we consider that the estimate of $\mu_y$ depends on the algorithm $f_i$ used to estimate the wake center position from the simulated flow field. To calculate the temporal variation of the wake deflection we divide the time series into shorter intervals $\Delta t$ and calculate the variance of this individual estimates about the mean.

## 2.2 Estimating the wake displacement by the change of wind direction

5 We consider the wind conditions at $x_1 = 2.5$ rotor diameters (D) upstream as reference inflow conditions to a wind turbine. This distance is chosen as the wind field closer to the turbine might be modified by the induction of the rotor (IEC-61400-12-1, 2005). More precisely our inflow information is hub height wind speed $u_h$ and wind direction $\alpha_h$ averaged at $x_1$ on a line extending $\Delta y = 2\,\mathrm{D}$ perpendicular to the expected mean wind direction (Fig. 1). We choose cross stream averaged variables instead of a point measurement as we consider them more representative for the wind conditions for the wind turbine rotor.

To estimate the wake displacement $y_0$ we assume an advection of the wake with the ambient wind. If the wind direction coincides with the $x$-axis ($\alpha_h = 0$), the wind flows along the $x$-axis and interacts with the wind turbine to form a wake structure that is advected downstream, supposedly centered around $y_0 = 0$. For wind directions $\alpha_h \neq 0$ the $x$-axis and wind direction differ and the center $\mu_\gamma$ of the wake is expected to be shifted by $y_0 = \Delta x_2 \tan \alpha_h$ along the $y$-axis (Fig. 1). As we only consider
deviations of the wind direction from the $x$-axis of less than 10 degrees, the change of $x_2$ with $\alpha_h$ is neglected.

This simple consideration already allows for a first estimation of how the uncertainty from the calculation of the wind direction can propagate into the error of the wake deflection estimation. For an error of the wind direction estimation of $\sigma_{\alpha_h} = \pm 5°(10°)$ the wake center displacement $y_0$ at $x_2 = 6\,\mathrm{D}$ downstream would have an uncertainty of $\sigma_{y_0} \approx \pm 0.5\,\mathrm{D}(1.0\,\mathrm{D})$.

## 2.3 Estimation of the wake center

Three different methods to estimate the wake center position are compared in this study to assess the bias introduced to $\mu_y$ by the choice of the method $f_i$. As a first approach the position of the wake is calculated by fitting the mean wake deficit at hub height to a Gaussian-like function.

$$f_h(y) = u_a exp\left(-\frac{(y-\mu_y)^2}{2\sigma_y^2}\right) \tag{4}$$

The center $\mu_y$ of the Gaussian is considered as the horizontal wake center, the amplitude $u_a$ as the wake deficit and $\sigma_y$ as a measure of the width of the wake.

As we have also information about the vertical structure of the wake, a 2D Gaussian-like fit as proposed by Trujillo et al. (2011) is used as alternative fitting routine.

$$f_{2D}(y) = u_a exp\left[-\frac{1}{2(1-r^2)}\left(\frac{(y-\mu_y)^2}{\sigma_y^2} - \frac{2\rho(y-\mu_y)(z-\mu_z)}{\sigma_y^2\sigma_z^2} + \frac{(z-\mu_z)^2}{\sigma_z^2}\right)\right] \tag{5}$$

with $\mu_z$ the equivalent to $\mu_y$ on the vertical axis and $r^2 < 1$ a correlation factor. For a perfect circular shape of the wake $r = 0$, whereas for an elliptic wake shape $r \neq 0$. Both functions are fitted to the data through a least-squares approach.

We introduce a third method to determine the wake position based on the available mean specific power in the wind (AP).
As the main interest of wind farm control is the increase of the power output of downstream turbines, we consider the position along the $y$-axis of a hypothetical turbine placed at $x_2$ that feels the lowest AP as the center point of the wake. For this purpose the cube of the mean flow in wind direction is averaged on circular planes of diameter D centered around hub height $z_h$. The AP is normalized by the air density, as density variations are not considered.

$$f_{\mathrm{AP}}(y) = 1/2 \int\limits_{y1}^{y2} \int\limits_{z1}^{z2} u^3(y',z')dz'dy', \ (y'-y)^2 + (z'-z_h)^2 \leq (D/2)^2 \tag{6}$$

The wake center $\mu_y$ is the value of $y$ that minimizes Eq. 6.

## 2.4  Temporal averaging interval

To study the uncertainty of the wake deflection by the used temporal averaging interval, we divide time series of inflow at $x_1$ and wake flow at $x_2$ in multiple time intervals $\Delta t$. We chose time intervals of respectively $\Delta t = 10, 3$ and $1\,\mathrm{min}$ as we consider them realistic for wind farm control.

For small $\Delta t$ the wind conditions at $x_1$ and $x_2$ become more and more uncorrelated, thus the advection time of the turbulent structures between these points is considered for each averaging interval. Turbulent structures in the wind field are expected to be transported by the mean wind following Taylor's hypothesis of frozen turbulence. To describe the time $\tau$ it takes for a structure to be advected from the position $x_1$ to the position $x_2$ we use the following approximation:

$$\tau = (\Delta x_1 + \Delta x_2)/u_h \tag{7}$$

with $\Delta x_1$ and $\Delta x_2$ being the distances from $x_1$ and $x_2$ to the wind turbine, respectively. In presence of a turbulent structure of lower velocity like a wind turbine wake, the advection velocity downstream of the turbine along $\Delta x_2$ is not well studied. Following Larsen et al. (2008) we assume that the wake is moved like a passive tracer by the ambient wind field. Thus the advection velocity downstream of the turbine remains the same as upstream.

Combining the methods presented in previous subsections we find multiple estimates of the wake deflection $\Delta y_\gamma$ by calculating the wind direction $\alpha_h$ and the wake center $\mu_y$ for different averaging intervals $\Delta t$, with the time series at $x_2$ shifted by $\tau$, and for different methods $f_i$ to identify the wake center from the wake flow.

## 2.5 LES model

The simulations presented in here are conducted with the LES model PALM (Maronga et al., 2015). PALM is an open source LES code that was developed for atmospheric and oceanic flows and is optimized for massively parallel computer architectures. It uses central differences to discretize the non-hydrostatic incompressible Boussinesq approximation of the Navier-Stokes equations on a uniformly spaced Cartesian grid. PALM allows for a variety of schemes to solve the discretized equations.

The following schemes are used in this study: Advection terms are solved by a fifth-order Wicker-Skamarock scheme, for the time integration a third-order Runge-Kutta scheme is applied. For cyclic horizontal boundary conditions a FFT solver of the Poisson equation is used to ensure incompressibility, while for non-cyclic horizontal boundary conditions an iterative multi-grid scheme is utilized. A modified Smagorinsky sub grid scale parametrization by Deardorff (1980) is used to model the impact of turbulence of scales smaller than the model grid length on the resolved turbulence. Roughness lengths for momentum and heat are prescribed to calculate momentum and heat fluxes at the lowest grid level following Monin-Obukhov similarity theory.

The simulations in PALM are initialized with a laminar flow field. Random perturbations of the flow during the start of the simulation initiate the development of turbulence. The statistics of the steady turbulence that develops after some spin-up time depend on the initial conditions provided for the fluid, e.g. the temperature profile, and the boundary conditions during the simulation, e.g. surface heat fluxes. For more information about the general capabilities of the model the reader is referred to Maronga et al. (2015).

## 2.6 Wind turbine model

The effect of the wind turbine on the flow is parameterized by means of an enhanced actuator disk model with rotation (ADM-R) as in Witha et al. (2014); Dörenkämper et al. (2015b). The rotor disk is divided into rotor annulus segments with changing blade properties along the radial axis. The blade segments positions are fixed in time but each owns an azimuthal velocity due to the clockwise rotation of the rotor. Local velocities at the segment positions are used in combination with the local lift and drag coefficients of the blade to calculate lift and drag forces. The forces are scaled for a three bladed turbine and are afterwards projected onto the grid of the LES by a smearing function with a Gaussian kernel as described in Dörenkämper et al. (2015b). In internal sensitivity studies we found that a value of twice the grid size is a good choice for the regularization parameter as also concluded by Troldborg et al. (2014). The rotor can be rotated around the $y$-axis and the $z$-axis enabling a free choice of yaw and tilt configuration. The influence of tower and nacelle on the flow is represented by constant drag coefficients.

The blade properties as well as the hub height of $z_h = 90\,\mathrm{m}$ and the rotor diameter of $\mathrm{D} = 126\,\mathrm{m}$ originate from the NREL 5MW research turbine (Jonkman et al., 2009). A variable-speed generator-torque controller is implemented in the same way as

described in Jonkman et al. (2009). Note that no vertical tilt is applied to the rotor to exclude the wake displacement that might result from a mean vertical momentum of the wake.

## 2.7 Precursor simulations

Precursor simulations of the atmospheric boundary layer for the representation of three different atmospheric stabilities, stable, neutral and convective, are conducted with the goal of creating different shear and turbulence characteristics but with the same mean wind speed and direction at hub height. All domains have a horizontal and vertical grid resolution of $\Delta = 5\,\mathrm{m}$ up until the initial height of the boundary layer in each simulation. Above this height the vertical grid size increases by $6\%$ per vertical grid cell. The roughness length is kept constant in all simulations at $z_0 = 0.1\,\mathrm{m}$, representing a low onshore roughness representative for low crops and few larger objects. The Coriolis parameter corresponds to $54°\mathrm{N}$. Cyclic lateral boundary conditions are used and the simulations are initialized with a vertically constant geostrophic wind. Due to Coriolis forces, bottom friction and stratification, height dependent wind speed and wind direction profiles evolve after several hours of spin-up time.

For the generation of a SBL, a constant cooling of the lowest grid cells is prescribed. The initial temperature profile of the potential temperature $\Theta$ and the rate of bottom cooling ( $d\Theta/dt = 1\,\mathrm{K}/4\,\mathrm{h}$ ) are set as in Beare and Macvean (2004). A CBL is established by prescribing a constant kinematic sensible heat flux of $60\,\mathrm{Wm}^{-2}$ at the bottom boundary. The bottom heat flux is fixed to zero for the NBL. The initial potential temperature profiles of the NBL and CBL are constant up to $500\,\mathrm{m}$ height with a strong inversion of $d\Theta/dz = 8\,\mathrm{K}/100\,\mathrm{m}$ between $500\,\mathrm{m}$ and $600\,\mathrm{m}$ and a stable stratification of $d\Theta/dz = 1\,\mathrm{K}/100\,\mathrm{m}$ up to the upper model boundary.

The results of the precursor simulations are shown in Fig. 3, Fig. 4 and Table 1. The simulations differ in their horizontal and vertical extent (see Table 1), a consequence of the different heights of the mixing layers and the different sizes of the largest eddies that need to be explicitly resolved. These simulations are afterwards used as initial wind fields for the main simulations described in Sect. 2.8 that include the impact of the wind turbine on the flow by the ADM-R parametrization. As intended, the domain averaged profiles have similar mean wind speed and direction at hub height but differ in vertical shear of the wind speed, wind veer and turbulence intensity (Fig. 3). The SBL is characterized by a strong vertical shear of wind speed and wind direction over the height of the rotor. Shear coefficient $\alpha_s = 0.30$ and Monin-Obhukov length $L = 170\,\mathrm{m}$ correspond to a stable to highly stable stability class following Wharton and Lundquist (2012). The wind direction changes by $8°$ from the lower rotor tip to the upper rotor tip. Below the top of the SBL at around $z_i = 300\,\mathrm{m}$, the wind speed has a super-geostrophic maximum, an event called low level jet, that has been documented in measurements onshore as well as offshore (Smedman et al., 1996; Emeis, 2014; Dörenkämper et al., 2015a).

The NBL and the CBL exhibit only low vertical dependency of the wind vector above the lower rotor tip. Responsible for the low vertical wind speed gradient is the increased amount of turbulent kinetic energy that leads to a stronger mixing. The spectra of the three velocity components at hub height shown in Fig. 4 reveal that not only the total amount of turbulent kinetic

energy is larger in the neutral and convective case, but the most energetic motion also occurs on larger scales.

The CBL represents a rather moderate convective boundary layer with $L = -180\,\mathrm{m}$ and a ratio between the boundary layer height $z_i$ and $L$ of $z_i/L = -3.6$. Characteristic for such moderate convective boundary layers in flat terrain are large roll-vortices, whose axes of rotation are approximately aligned with the mean wind direction and that have a vertical extension up to the top of the boundary layer (Etling and Brown, 1993; Gryschka et al., 2008). The presence of these vortices can be seen in the highly energetic low frequently motion of the $v$- and $w$- components and the large variance of the wind direction.

The meteorological conditions of the CBL and SBL simulation cases are regularly occurring at wind farm sites (Hansen et al., 2012; Vanderwende and Lundquist, 2012; Wharton and Lundquist, 2012). Numerical simulations comparable to the CBL and NBL case are studied in Churchfield et al. (2012), while Mirocha et al. (2015) simulate even stronger stable and convective conditions, which are motivated by measured events.

## 2.8 Setup of the wind turbine wake simulations

For the main simulations a turbulence recycling method (Maronga et al., 2015) is used at the upstream domain boundary instead of a cyclic boundary (Fig. 2). This allows for studying a single turbine along the $x$-axis instead of an infinitively long row of turbines. Undisturbed outflow at the right boundary is ensured by a radiation boundary condition. For the use of the turbulent recycling method the model domain from the precursor simulations is extended along the x-axis and the recycling surface is positioned at the domain length $L_{xp}$ of the precursor run. Test simulations showed a minimum of $L_y^{min} \approx 8\,\mathrm{D}$ to prevent blockage of the flow by the turbine and a minimum distance between recycling surface and turbine of $L_I^{min} \approx 3\,\mathrm{D}$ to prevent an influence of the induction zone on the turbulence at the recycling surface.

The main simulations of the NBL and SBL are conducted for single turbines with a different yaw angle to the $x$-axis. For each change in yaw angle a separate simulation of $25\,\mathrm{min}$ length is conducted from which the first $5\,\mathrm{min}$, during which the wake still develops, are discarded from the analysis. Yaw angles ranging from $-30°$ to $30°$ in steps of $10°$ are chosen. Positive yaw angles are defined as a clockwise turning of the rotor when seen from above and the wind coming from the left hand side.

In the CBL the domain width $L_y$ is more than 6 times larger than the minimum size of $L_y^{min}$. We use this to include all different turbine yaw angle configurations in one simulation consisting of two staggered rows of four turbines each, separated by more than $L_y^{min}$ in y and $12\,\mathrm{D}$ in $x$-direction. The distances are chosen large enough that a mutual interaction of the turbines can be excluded. Each of the turbines had a different yaw angle to the $x$-axis and the simulation was run for 65 minutes from which the first $5\,\mathrm{min}$ were discarded. The longer simulation time of the CBL is motivated by the larger turbulence length scales of the flow that cause longer necessary averaging intervals to get information about mean properties. Note that due to the cyclic lateral boundary conditions, the turbines in all simulations are part of an infinite row along $y$ seperated by more than $L_y^{min}$.

## 3 Results

In this section we compare the results of the main simulations with presence of wind turbines. The vertical planes of the LES flow that are shown on the following pages represent the view of an upstream observer looking downstream. If not explicitly noted otherwise, the zero coordinate of the $x$-axis coincides with the $x$-position of the rotor center and the zero coordinate of the $y$-axis with $y_0$, i.e. the zero coordinate of $y$ corrected by the measured inflow wind direction $\alpha_h$. The $y$-axis is positive to the left hand side of the upstream observer.

### 3.1 Neutral atmospheric boundary layer

We start the analysis with the NBL, as this case is the most studied case in wind energy applications. Figures 5(a-c) show vertical planes of the wake deficit $u_{def}$, averaged over the whole simulation time, for three different yaw angles $\gamma$ at $x_2 = 6\,\mathrm{D}$. The velocity $u_{def}$ is defined as the difference between the inflow velocity profile of $u(y,z)$ measured as inflow at $x_1$ and averaged along $\Delta y = 2\,\mathrm{D}$ and the velocity field $u(y,z)$ at $x_2$ downstream of the wind turbines (Fig. 1). The isolines of the 2D fitting method $f_{2D}$ are denoted by dashed contours. The wake deflection $\Delta y_\gamma$ that results from this routine is visible as the innermost ring. Cross sections of Fig. 5(a-c) at hub height are shown together with the results of $f_h$ and $f_{AP}$ in Fig. 5(d). The wake centers are the positions along $y$ for which the functions take the smallest values.

As apparent in Fig. 5 the wake deficit is lower for the two cases of turbines with a large yaw angle, a consequence of the loss of energy yield and induction, if a wind turbine is yawed out of the wind direction. For a positive (negative) yaw angle the wake deficit is deflected to the left (right) when looking from upstream. Figure 6 shows the mean deflection $\Delta y_\gamma$ of the wake center for multiple distances downstream of the rotor using the three different approaches $f_i$. The Gaussian-like fit at hub height $f_h$ returns the largest deflection of the wake. The smallest deflection is found when the wake is approximated by the 2 D normal fit $f_{2D}$ while the wake position of minimal $f_{AP}$ lies mostly between the two curves.

The reason for the different output of the three methods is the deviation of the wake from a perfect symmetric shape as evident in Fig. 5. The crescent shapes of the wakes indicate that the lateral displacement is largest at the height around the rotor center while it is lower around the upper and the lower rotor tip, which explains the largest magnitude of wake deflection for $f_h$.

A look at the cross stream component of the flow reveals the origin of the crescent shape of the wakes of a yawed turbine. Figure 7 shows the residual cross stream component of the flow in the near wake. The residual component is the difference between the inflow profile and the downstream wind field. For $\gamma = 0°$, the dominant feature of the cross stream flow is the counterclockwise rotation of the wake that is induced by the clockwise rotation of the rotor. For $\gamma \neq 0°$, the rotation is superimposed by the induction of cross stream momentum caused by the yawed turbine. Figure 7 (a,c) show that this cross stream momentum is either opposing the rotor rotation below or above the hub, which, together with the influence of wind veer, leads to the asymmetries further downstream as evident in Fig. 5 (a,c).

As apparent in Fig. 7 the induced cross stream momentum also triggers a counter momentum above and below the rotor area. The opposing cross stream velocities appear to be responsible for the varying magnitude of lateral displacement at different heights and the crescent shape of the wake further downstream. The counter momentum is stronger below the rotor area,

which is likely to be related to the presence of the bottom just $27\,\mathrm{m}$ below the blade tip.

To assess the influence of the temporal averaging interval on the standard deviation of the wake deflection, $\Delta y_\gamma$ is calculated for different time intervals. Advection of frozen ambient turbulence between $x_1$ and $x_2$ is considered by shifting the second time interval by $\tau$ (Eq. 7). To have more than two estimates for the 10 min interval, the intervals are overlapping to a

large degree resulting in seven individual estimates per yaw configuration. Figure 8 shows the spread of the estimates of $f_{2D}$ at two different positions $x_2$. We find that the standard deviation of the wake deflection appears to be independent of the yaw angle but depends on the temporal averaging interval. The used fitting method has little influence on the standard deviation of the mean wake deflection in the NBL (Table 2).

### 3.2   Stable atmospheric boundary layer

As shown earlier in Fig. 3, the simulated SBL is characterized by lower TI and a stronger vertical shear of wind speed and direction than the NBL. For the simulated wind turbine wake in the SBL, the strong wind veer leads to a strong slanted shape of the wake deficit, even if the rotor plane is perpendicular to the wind direction at hub height (Fig. 9b). Below the rotor center, the wake is shifted towards the left hand side and above towards the right hand side. Thus, the extend of the wake cross section at hub height (Fig. 9d) is less representative for the whole wake extension than in the NBL simulation (Fig. 5). The amplitude at

$x_2 = 6\,\mathrm{D}$ of the wake deficit $u_{def}$ is larger than in the NBL. The larger amplitude can be related to the lower ambient turbulent kinetic energy and to the lower fluctuation of the inflow wind direction.

The wakes for $\gamma \neq 0°$ show a similar crescent shape to the wakes in the NBL. The differences between the deficit position at hub height and around the upper and lower rotor tips are even larger, a consequence of the addition of induced momentum

by the yawed turbine and ambient wind veer. In the case of a yaw angle of $\gamma \approx -30°$ the lower part of the wake detaches from the rest of the structure. In contrast to the fit $f_{2D}$ of the wake at $\gamma \approx 30°$ this detached part is neglected by the optimal fit.

The trajectories of the wake deflection shown in Fig. 10 have a distinct bias to the right of the rotor. This appears in all trajectories but is strongest in the $f_{2D}$ trajectory where basically no deflection to the left is found. The wake deflection to the

right may be related to two different mechanisms. Firstly, it can be related to advection of lower momentum from below the rotor to one side and advection of high momentum from above the rotor to the other side of the wake by its rotation. The second effect that could be responsible for the deflection of the wake to the right is the stronger veer of the wind in the upper rotor half, where the mean flow is towards the right, compared to the lower rotor half, where the mean flow is slightly towards the left. Trajectories of simulations with a reversed rotation of the rotor show that the sense of rotation is not exclusively responsible

for the bias to the right as this would lead to a mirroring of the trajectories about the wind direction for opposite rotor rotations (Fig. 10). As apparent in Fig. 9, the wake center is located a little higher than hub height, therefore the ambient wind direction at wake center height could also lead to a slight advection towards the right. Thus both effects seem to be responsible for the difference between the wake deflection in the SBL and the NBL.

The uncertainty of the estimate of the wake deflection is much smaller in the SBL than in the NBL for all time intervals (Fig. 11). Compared to the NBL, the variance of the wind direction (Fig. 3b) is lower and the energy of the cross stream motion (Fig. 4) is already low on the minute scale. Thus, a $1\,\mathrm{min}$ averaging window filters most of the cross stream fluctuation that might be responsible for the uncertainty of the prediction of the flow field between $x_1$ and $x_2$ and therefore the uncertainty of

10 the wake deflection.

### 3.3 Convective atmospheric boundary layer

The deflected wakes in the CBL show a completely different behavior than in the previous presented boundary layer simulations. Figure 12 shows the $yz$-transects as in Fig. 5 and Fig. 9 but for the CBL. The results are averaged over one hour of simulation time instead over $20\,\mathrm{min}$ like in the other simulations. The large deficit width in Fig. 12 is mainly a consequence

of the large variance of wind direction (Fig. 3(b)) during the averaging time interval, that leads to a strong fluctuation of the wake position (Larsen et al., 2015; Machefaux et al., 2015a). A consequence is a much weaker mean deficit than in the NBL and SBL simulations.

As Fig. 12 shows, the wake deflection to the left (right) for a positive (negative) yaw angle is not found in the results of

20 the CBL simulation. This does not only hold for the long time average but also for shorter time intervals $\Delta t$ as apparent in Fig. 13. The uncertainty of the estimated wake deflection is less dependent on the averaging interval than in the other simulation (Table 2).

Following the considerations made in Sect. 2.3 about the uncertainty of the wake deflection due to the uncertainty of the

25 wind direction, an approximate error of $\pm 2.5°$ of the $3\,\mathrm{min}$ wind direction $\alpha_h$ can be derived from the spread of the $3\,\mathrm{min}$ results (Table 2).

A large spread of yaw angles of the turbines to the wind is encountered during the simulation (Fig. 13). The reason for the spread are the wide streaks of the convection rolls that create strong cross stream components (Fig. 14) , a feature that dis-

30 tinguishs the CBL from the other simulated cases. Due to this feature, the local inflow wind direction usually differs from the domain-averaged wind direction, shown in Fig. 3, to which the turbines are originally yawed. These streaks explain the spread of identified wind directions but can not explain the high variance of the wake deflection for the same yaw and inflow angle. Moreover, the averaged wind speed and direction measured in front of the turbine appears to be insufficient to characterize the flow further downstream.

To test the similarity of the free stream flow at different streamwise locations we calculate the root mean square error ($RMSE$) of two time-series in undisturbed flow with and without considering the time shift $\tau$ (Fig. 15). Wind speed and wind direction are averaged at hub height along a cross stream distance as described in Sect. 2.3. A shift of the downstream time series by $\tau$ has the largest effect on the similarity of the wind conditions in the CBL, where especially the variance of the wind direction is large. On the other hand that means that a bad estimation of $\tau$ introduces the largest error to the estimation of $y_0$ in the CBL.

## 4   Discussion of the wake deflection estimation

Three different sources of uncertainty of the wake deflection estimation are evaluated in this study. First we show that the incoming wind shear and veer has to be well known by comparing the results from the neutral and stable thermal stability situation. The influence of shear and veer is not considered yet by studies of potential improvement of the wind farm efficiency with wind farm control like Annoni et al. (2015) and Gebraad et al. (2016). Table 3 shows the coefficients derived from the two simulations for the analytical description proposed in Jimenez et al. (2010) and Gebraad et al. (2016) compared to their results. Gebraad et al. (2016) show that the energy yield of a small wind farm can be well predicted by a simplified parametric model, which is fitted to simulated atmospheric conditions of neutral stability, and that the energy yield of a small wind farm can be improved by more than 10 percent for certain scenarios. Assuming the same parameters for the stable wind field from our study would lead to a miscalculation of the wake position which corresponds to a yaw induced deflection by a yaw angle of about $10°$. Thus, the described parametrization of the model would likely propose an unfavorable control for stable situations. A proper description of the wake trajectory in stable situations is important as the interest to apply wind farm control in stable atmospheric stability should be higher than in more turbulent conditions due to the increased wake losses. With the high occurrence of stable situations onshore (Vanderwende and Lundquist, 2012; Wharton and Lundquist, 2012) as well as offshore (Barthelmie and Jensen, 2010; Dörenkämper et al., 2014) the difference in the wake trajectory might be even worth considering in the design process of a wind farm.

As a second source of uncertainty we consider the choice of the method to derive the wake position. These methods are most often dependent on the measurement device thus we do not expect that it will be possible to establish a universally applicable method in the near future. For future studies aiming to study the deflection of the wake we emphasize that the choice of wake fitting routine for the measured wind field has significant influence on the results in particular when the turbine yaw angle is large.

The third source of uncertainty that is considered in this study is the influence of the time averaging interval to find the wake deflection. The underlying question behind this analysis is: At what time scales makes wind farm control sense and what needs to be taken into account at the different time scales. In the NBL and SBL cases the estimation of the wake deflection on a $10\,\text{min}$ scale shows only little variance. However, here we benefit from the steady wind field in the LES where we don't expect a change of wind direction over this time interval. In practice, meso-scale wind fluctuations might cause a change of the

wind direction on this time scale. For smaller time intervals than 10 min the variance of the wake deflection increases, thus a prediction of the wake position by measuring the inflow becomes more uncertain.

The CBL analysis differs from the two other cases as we find no correlation between yaw angle of the turbine and wake deflection on any of the tested time averaging intervals. This makes a prediction of the wake position more uncertain and makes an interference by yaw control unreasonable. Apparently, the stochastic fluctuation of the wake caused by the large fluctuations of the cross stream component are superimposing the trajectory change of the wake caused by the induction of the turbine to a degree that the latter signal is not detectable any more. The larger fluctuation of the wake trajectory in convective conditions has been shown before in measurements and simulations (Keck et al., 2014; Mirocha et al., 2015) but has not been related to the applicability of wind farm control, yet.

Investigating the hypothesis of frozen turbulence in flow undisturbed by the wind turbine shows that the consideration of the time delay between the time series at two streamwise positioned measurements is especially important in the CBL. However, in flow with a wake structure of lower mean velocity than the ambient wind field, the advection velocity relevant for the lateral movement of the structure is not well-defined. Thus, the time delay between inflow measurement and wake measurement can not be estimated accurately. A better understanding of the relevant advection velocity of the wake might improve a prediction of the wake position in highly turbulent environments. Attempts to improve the description of the advection velocity are made for example in Machefaux et al. (2015b).

A source that we do not address in this study is the uncertainty of the wind direction estimate by the error of the measurement device that is used. The cross stream average of hub height flow upstream of the turbine, that we use here, is just one possibility to measure the inflow. The only way to apply this method in the field would be by using nacelle based lidar systems like proposed in Schlipf et al. (2013).

The shown simulations represent only examples of thermal stability conditions for stationary and barotropic flow. In addition to atmospheric stability other factor like baroclinicity and topography influence the wind profile. Thus, from the shown simulations we can conclude little about the influence of atmospheric stability at a specific location. For the fine-tuning of wake models it would be beneficial to study the exact effect of shear and veer on the wake position and shape in more detail.

## 5  Conclusions

In this study we contribute to the current discussion about wind farm control by considering atmospheric stability and uncertainty of the wake deflection estimation. From LES case studies of yawed wind turbines in atmospheric boundary layers of different thermal stratification we conclude that both a precise wind direction measurement and measurements of shear and turbulence of the flow are necessary to be able to accurately predict the position of the wake downstream of the turbine. These

factors should be considered by any comprehensive study aiming to evaluate the costs and benefits of wind farm control concepts. As current approaches of wind farm control require a loss of power as well as often an increased structural load at upwind turbines, a wrong prediction of the wake position will most likely not lead to an improvement of wind farm performance.

We also emphasize that the wake position in a turbulent atmospheric boundary layer becomes more and more stochastic for small time intervals. Furthermore, in a highly turbulent environment, the use of yawed turbines to deflect the wake might even not be reasonable at all as we find no correlation between the wake position and the turbine yaw angle relative to the measured inflow in a simulation of a convective situation. However, the use of wind farm control is regarded to produce the strongest improvement of wind farm performance in stable conditions because the power losses due to wakes are highest. Our

study shows that an application of an intentional wake deflection in these conditions might be feasible if the trajectory is well described because the fluctuation of the wake position is low.

*Acknowledgements.* The work presented in this study has been done within the national research project "CompactWind" (FKZ 0325492B) funded by the Federal Ministry for Economic Affairs and Energy (BMWi). Computer resources have been partly provided by the North German Supercomputing Alliance (HLRN) and by the national research project "Parallelrechner-Cluster für CFD und WEA-Modellierung"

(FKZ 0325220) funded by the Federal Ministry for Economic Affairs and Energy (BMWi). The authors further want to thank D.Bastine and B.Schyska for valuable discussions about the content of the manuscript.

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

| | Setup | | | | | | Results | | | | | |
|---|---|---|---|---|---|---|---|---|---|---|---|---|
| | $L_x$ | $L_{xp}$ | $L_y$ | $L_I$ | $L_z$ | $n_T$ | $u_h$ | $TI_h$ | $\alpha_s$ | $\delta\alpha$ | $L$ | $z_i$ |
| | [D] | [D] | [D] | [D] | [D] | | [ms$^{-1}$] | [%] | [] | [°] | [m] | [m] |
| SBL | 30.5 | 11.4 | 7.6 | 3.0 | 4.5 | 1 | 8.4 | 4.0 | 0.30 | 8.2 | 170 | 300 |
| NBL | 61.0 | 23.7 | 20.3 | 6.0 | 13.6 | 1 | 8.3 | 8.3 | 0.17 | 2.2 | $\infty$ | 550 |
| CBL | 132.0 | 81.3 | 50.8 | 8.0 / 20.0 | 11.6 | 8 | 7.8 | 13.3 | 0.08 | 0.6 | -180 | 650 |

**Table 1.** Setup of the three simulations and results by the end of the prerun. Domain dimensions (see Fig. 2) are given in multiples of rotor diameter D. The number of turbines in the main simulation is $n_T$. Results consist of wind speed $u_h$ and turbulence intensity $TI_h$ at hub height, wind shear coefficient $\alpha_s$ and veer $\delta\alpha$, both evaluated between lower and upper rotor tip, Monin-Obukhov-Length $L$, and boundary layer height $z_i$.

| | std($f_h$) | | | std($f_{2D}$) | | | std($f_{AP}$) | | |
|---|---|---|---|---|---|---|---|---|---|
| | [$10^{-1}$D] | | | [$10^{-1}$D] | | | [$10^{-1}$D] | | |
| $\Delta t$ | 10 | 3 | 1 | 10 | 3 | 1 | 10 | 3 | 1 |
| SBL | 0.1 | 0.3 | 0.5 | 0.1 | 0.3 | 0.5 | 0.1 | 0.3 | 0.5 |
| NBL | 0.4 | 1.2 | 2.2 | 0.4 | 1.3 | 2.2 | 0.3 | 0.7 | 1.6 |
| CBL | 1.4 | 2.4 | 2.8 | 1.3 | 2.4 | 3.0 | 2.0 | 2.2 | 2.3 |

**Table 2.** Standard deviation of the wake deflection at $x_2 = 6\,$D for different $\Delta t$[min]. Values are averages over all seven yaw configurations. Note that the 10 min standard deviation might be biased as the intervals are not strictly independent.

| | $f_h$ | | $f_{2D}$ | | $f_{AP}$ | |
|---|---|---|---|---|---|---|
| | $k_d$ | $a_d,b_d$ | $k_d$ | $a_d,b_d$ | $k_d$ | $a_d,b_d$ |
| SBL | 0.14 | -7.7,-1.4 | 0.23 | -8.1,-2.1 | 0.19 | -6.0,-2.4 |
| NBL | 0.16 | -3.1,0.4 | 0.25 | -2.8,0.9 | 0.18 | -2.4,0.3 |
| Jim. | 0.06 | - | - | - | - | - |
| Geb. | 0.15 | -4.5,-1.3 | - | - | - | - |

**Table 3.** Best fit parameters to the wake deflection output of the different methods using Gebraad et al. (2016), Eq.(12). Comparison with the results of the aforementioned study and with Jimenez et al. (2010). The parameter $k_d$ defines the recovery of the wake trajectory to the mean wind direction, and $a_d$ and $b_d$ the displacement due to the interaction of wind shear and rotation of the wake.

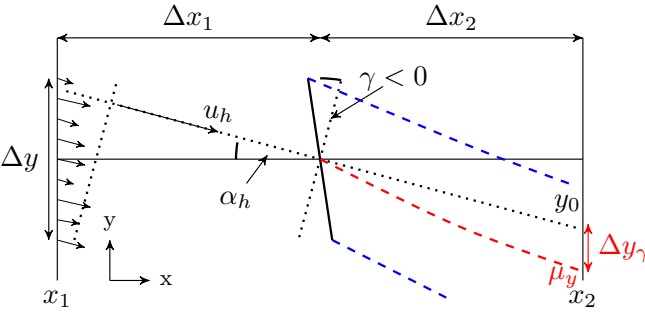

**Figure 1.** Conceptual image of the method to calculate the wake deflection $\Delta y_\gamma(x_2)$ by using the inflow wind direction $\alpha_h(x_1)$ of the wind speed $u_h(x_1)$ at hub height and the position of the wake center $\mu_y(x_2)$. Here, the x-axis is the mean wind direction. The yaw angle $\gamma$ is defined relative to $\alpha_h$, with $\gamma > 0$ for a clockwise turning of the rotor. Inflow wind speed and direction are averaged along $\Delta y$.

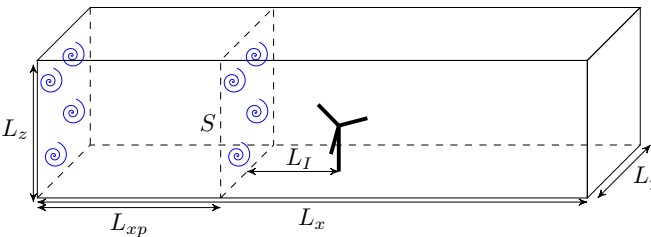

**Figure 2.** Domain of the main simulations. $L_{xp}$ is the length of the prerun domain. The turbulence at the recycling surface $S$ is used as input at the inflow again. $L_I$ is the distance from the recycling surface to the wind turbine.

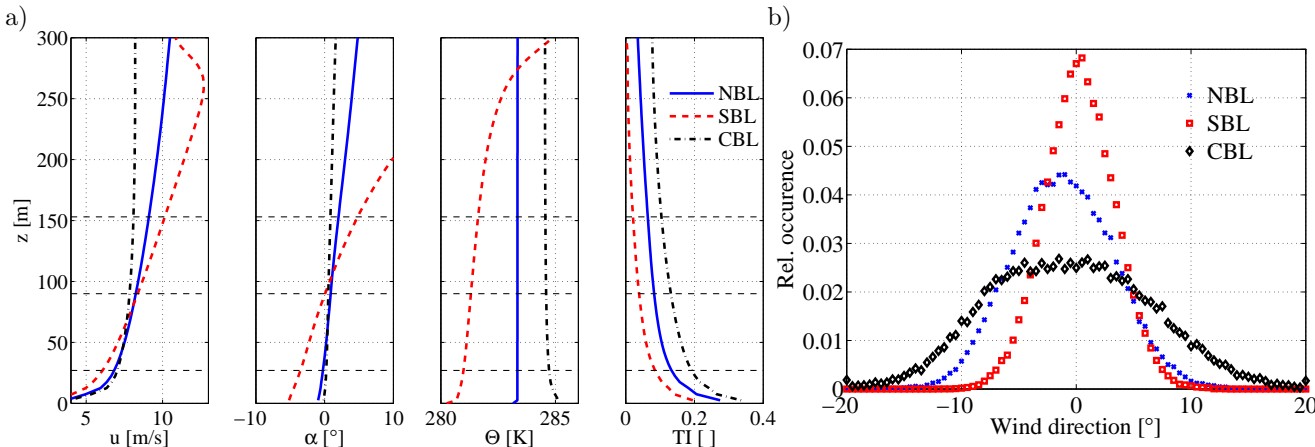

**Figure 3.** Flow statistics during the last hour of the precursor simulations. (a) Horizontally averaged vertical profiles of wind speed, flow direction, potential temperature and turbulence intensity. Horizontal lines denote the height of the blade tips and the hub. (b) Distribution of the 1 Hz wind direction from point measurements at hub height.

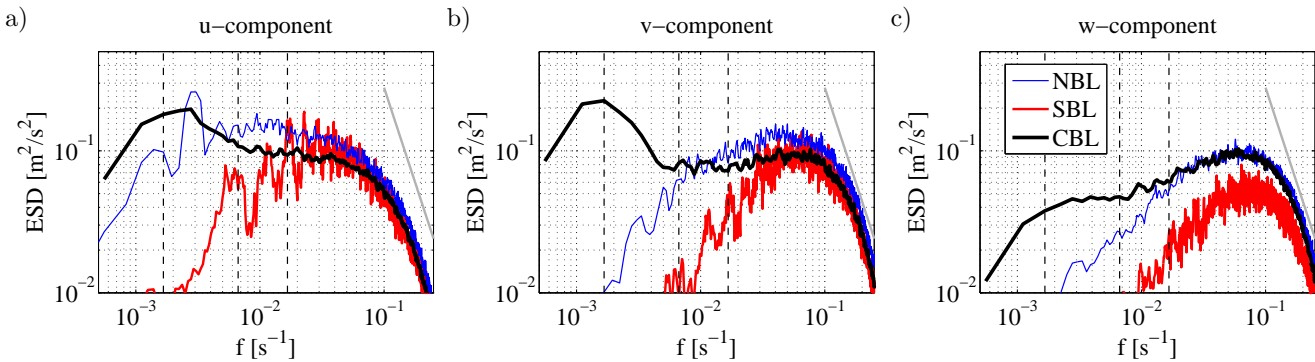

**Figure 4.** Energy spectral density of the three different wind components at hub height during the last hour of the precursor simulations. The gray line denotes the slope of the Kolmogorov cascade. Vertical lines are at T = 10 min, 3 min and 1 min.

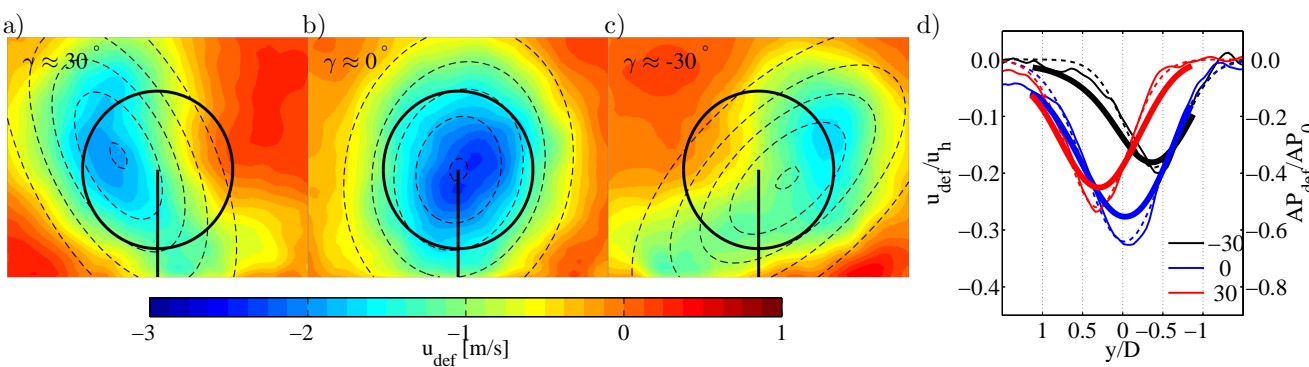

**Figure 5.** (a-c) Mean wake deficit 6 D downstream of a wind turbine in the NBL. The turbine is yawed by (a) $30°$, (b) $0°$ and (c) $-30°$. Straight contours denote the position of the upstream turbine. Dashed contours are the isolines of constant $f_{2D}$. (d) Cross sections of normalized $u_{def}$ at hub height (thin) and results of $f_{AP}$ (bold) and $f_h$ (dashed).

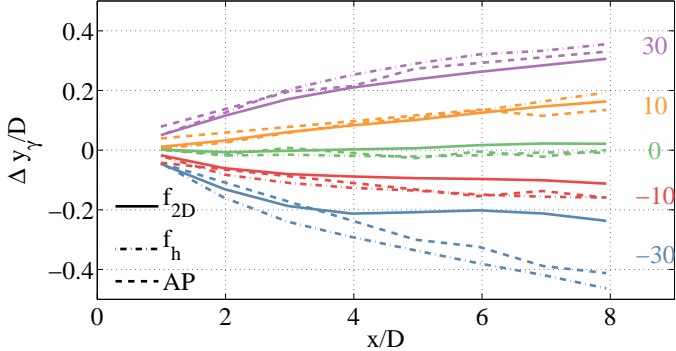

**Figure 6.** Wake deflection trajectories in the NBL from different fits to the data. Numbers on the right denote the turbine yaw angle for the different trajectories.

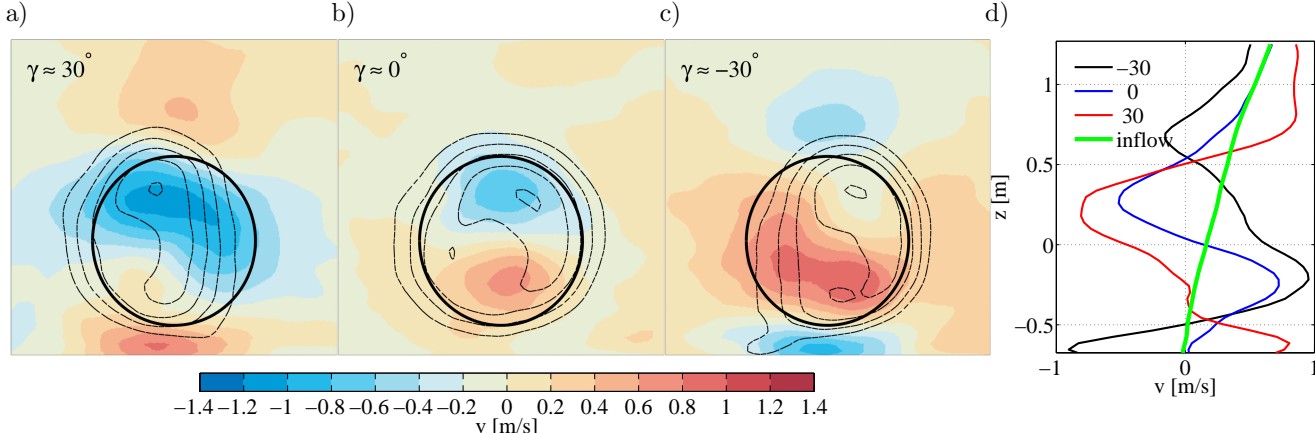

**Figure 7.** (a-c) Residual cross stream component of the flow at $x_2 = 2\,\mathrm{D}$ downstream of the wind turbine for the same simulations as in Fig. 5. Positive (negative) values stand for a flow to the right (left). Dashed contours denote the position of the wake deficit. (d) Vertical profile of the total $v$-component at $y_0$ and $x_2$, and the average inflow profile.

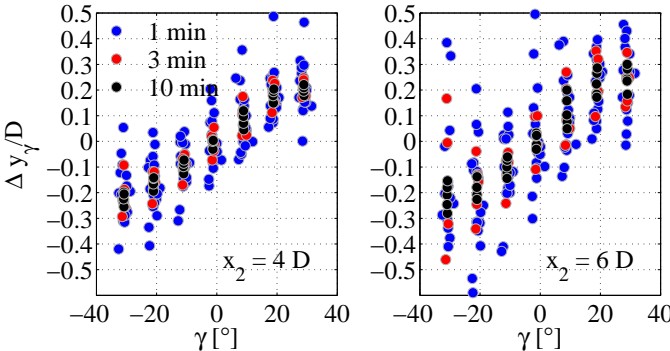

**Figure 8.** Scatter plot of the horizontal wake deflection in the NBL from the $f_{2D}$-fit over yaw angle $\gamma$ at different downstream positions $x_2$ and for different averaging intervals.

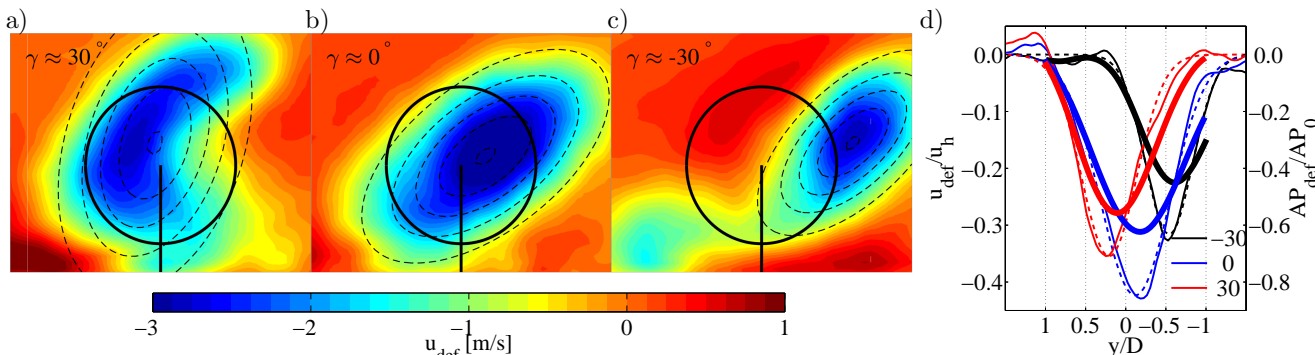

**Figure 9.** Same as in Fig. 5 but for the SBL simulation.

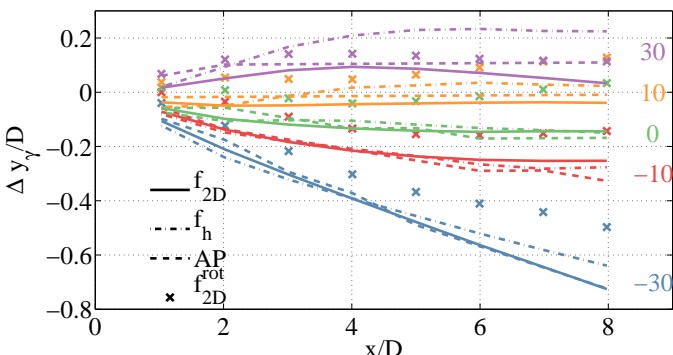

**Figure 10.** Same as in Fig. 6 but for the SBL simulation. Crosses mark the wake trajectories for simulations with opposite sense of rotation of the rotor.

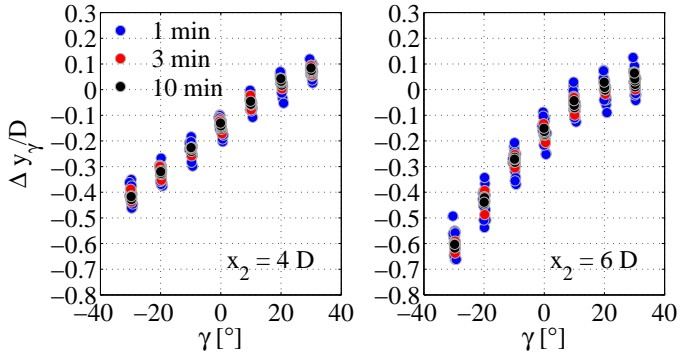

**Figure 11.** Same as in Fig. 8 but for the SBL simulation.

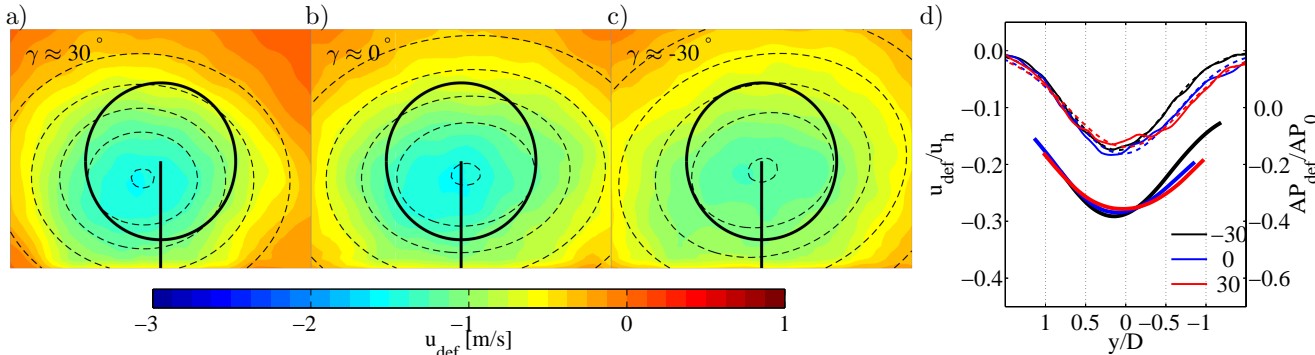

**Figure 12.** Same as in Fig. 5 but for the CBL simulation and for a time series of 60 min.

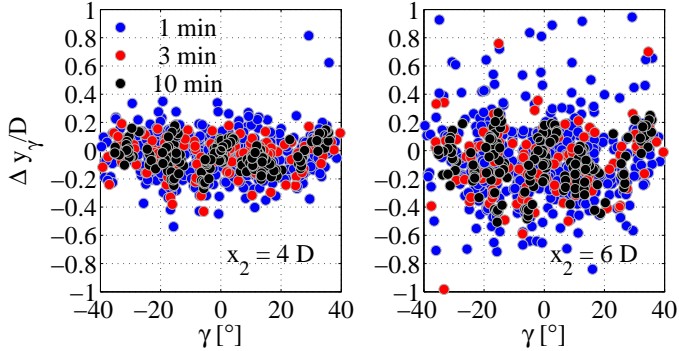

**Figure 13.** Same as in Fig. 8 but for the CBL simulation and for a time series of 60 min.

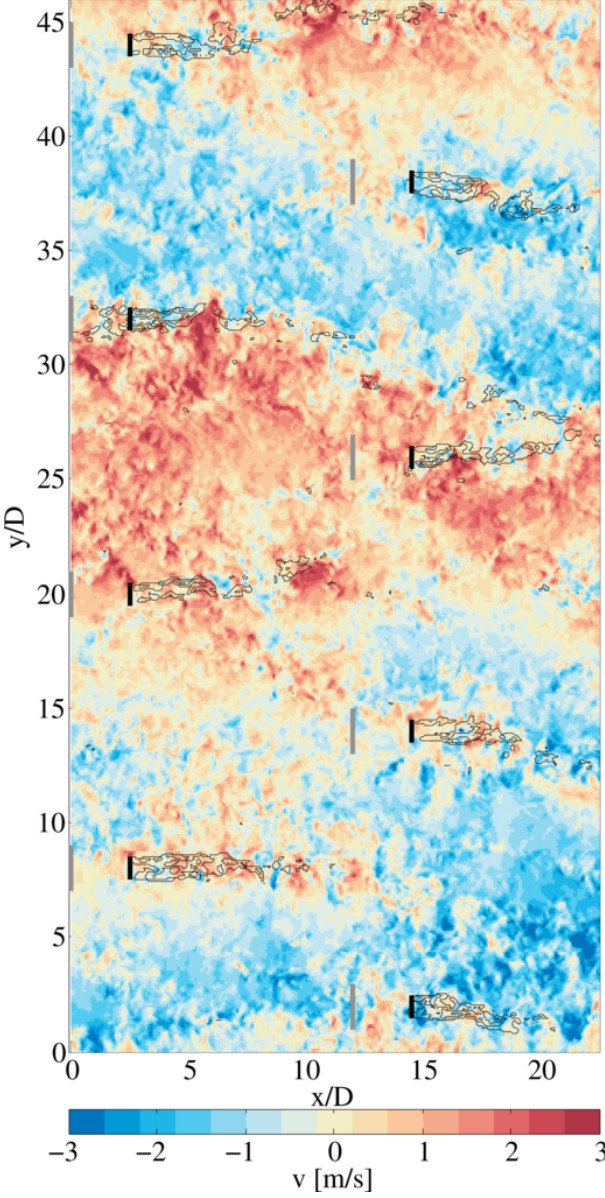

**Figure 14.** Example of the instantaneous $v$-component at hub height in the CBL. Turbine wakes are denoted by black contours. Black lines denote the rotor positions, gray lines denote the position of the inflow measurement for each turbine.

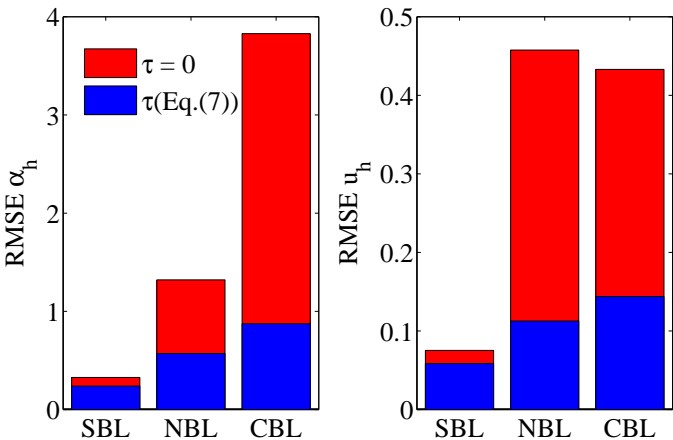

**Figure 15.** RMSE of the time series of $3\,\mathrm{min}$ averaged $a_h$ and $u_h$ at two different positions in the model domain separated by $\Delta x = 8\,\mathrm{D}$, with a advection time shift of the downstream time series of $\tau$ (Eq. 7) and without time shift ($\tau = 0$).