# Peer review of "Estimating the wake deflection downstream of a wind turbine in different atmospheric stabilities: An LES study"

_Wind Energy Science, 2016_

## Referee Comment (RC1) · Anonymous Referee #1 · 6 Jun 2016

Review for WES of
**Estimating the wake deflection downstream of a wind turbine in different atmospheric stabilities.**
By Vollmer et al

The paper sets out to address by means of LES the effect of wind turbine yaw on the deflection of a turbine wake in three types of wind: neutral, stable and convective. The deliberate deflection of wakes as a means of reducing adverse affects on downwind turbines is of practical interest. The paper presents interesting results which deserve to be published once some improvements to the manuscript are made, as outlined below.

Authors and affiliations - why superscript 1 in each?
L9 "Uncertainty in the"
L11 'increases with decreasing stability' Does this mean 'increases with increasing stability (from unstable to stable)'?
L17 "can have a significant .. 10-15 rotor diameters or more downstream"
L23 'The latter'

L18 'even suggested considering' But, what is the sentence saying anyway?
L30 'influence of atmos stability on the control mechanisms themselves' This seems an odd way of putting it. - the issues is how the control would need to change with the change of wind conditions as influenced by stability. There is no influence on the control mechanisms as such.

This brief review refers to stability effects as observed from field measurements and computations. There has also been recent work via laboratory stratified flow wind tunnel experiments - e.g. in Boundary-Layer Met.: Chamorro, Hancock and co-workers. On meandering there has been work by Aubrun and co-workers in J Wind Engin and Ind Aerodyn. These could be mentioned too.

Figure 1 is confusing. With yaw there is only one change; the hub axis with respect to a reference mean wind direction - at say hub height. Fig 1a is sufficient, Fig 1b is not needed and adds confusion. Fig 1a is sufficient when there is wind veer. Also, the yaw as shown in the fig is negative. Perhaps this should be mentioned in the figure caption, or the figure redrawn with positive yaw illustrated. (Is there consistency in sign of yaw angle as used later in the paper?)

Section 2.1
This seems unnecessarily complicated. At any instant, t, there will be a velocity profile $U(t, y)$ at given x. Over a short time internal $\Delta t$, there will be an short-period averaged profile $U'(t, \Delta t, y))$, which will vary with t, except when $\Delta t$ is very long in which case the profile will no longer be a function of t. $U'(t, \Delta t, y))$, for given t, will vary with x, and so can be used to define the wake (for that t and $\Delta t$). The long-term average – ie large $\Delta t$ – will define a mean

position varying with x. The short-term average will be a variation about this, but it does not make sense to refer to this as an error. The variation about the steady state is important is important as far as any downwind turbine is concerned.

Section 2.3
It seems slightly odd to use wake profile forms that are 'symmetrical' when the results, as might be anticipated (and are) clearly not symmetrical. Could other measures be used? Short period averaged-centre as defined by a velocity minimum, and wake half width?

Section 2.4
L11. With no wind turbine present there may be a reasonable strong correlation between the x1 and x2 stations – Taylor's hypothesis would be approximately valid. But a turbine largely destroys the correlation.

L 11 Why mention of humidity? This is an added complication if this is being modelled too.

Section 2.7
Mesh size of 5m seems very large. Surely, this should say, $\Delta x$, $\Delta y$, $\Delta z$, rather than just $\Delta x$?

Section 2.8
L31. Shouldn't the points made here be made in section 2.7?

Fig 3. It would be very useful to have the temperature profiles in this figure. This should be included as it is basic to non-neutral flow. (Suggest use I rather than TI; I is commonly used in meteorology.) It would also be helpful to include the cross-flow as an angle.

L17 does this comment not apply to CBL as well as SBL and NBL?
L20 'of an upstream observer looking downstream.' is helpful to be quite clear.

Fig 5. Presumably, the differences between a) and c) not being anti-symmetric is rotation in the wake. Which way is the turbine rotating, and therefore the wake in the opposite direction? Does this assist or oppose wind veer? I don't think there is mention of this.

Figs 5, 9 and 12. Say or show which way the rotor is rotating.

Fig 6. Why not show these as lines rather than points - perhaps using solid, dashed and broken lines?
L17. 'The resulting votices..' How does the vorticity get there, or is it an irrotational rotating motion?
L26 "The fitting method used.."
L32 ".. the extent.."

L5 – Paragraph. Isn't it just that the turbulence intensity is larger in the CBL case? Are these large scale structures not seen in the NBL and SBL to any degree?

L10 paragraph. This seems to overlook the much large v at low frequency, as shown in figure 4.

L25 '.. measurement device used.'

Table 3. Parameters are not introduced, defined and discussed in the text.

Fig 14. y/D and x/D, not as shown

---

## Referee Comment (RC2) · Anonymous Referee #2 · 13 Jul 2016

Intentional yaw misalignment of wind turbines as a way to control/optimize wind farm performance is indeed a relevant scientific topic. One part of this question is the potential for increased power production – another aspect is the increased loading following from WT yaw operation, which in principle is a cost. This paper links to the first mentioned aspect only and discusses whether or not the required (precise) knowledge of downstream wake trajectories can be achieved. This is crucial for the relevance of wind farm yaw control. The paper ends by concluding that for highly convective flow conditions, wind farm yaw control cannot be recommended, whereas the approach might be feasible under SBL conditions. However, there is no firm conclusion on the perspectives for the yaw-control approach under NBL conditions

– i.e. whether sufficient precise knowledge of downstream wake trajectories is likely achievable in practice. The scientific approach is sound, but the representativeness of the selected SBL/CBL cases should be discussed. References to relevant related work are in general good, but a few additional references should be added. The suggested references are found in the attached detailed pdf-review at the relevant passages in the manuscript. The detailed review moreover indicates a few misprints, asks for a few additional discussions/clarifications and suggests some editorial modifications.

Please also note the supplement to this comment:
http://www.wind-energ-sci-discuss.net/wes-2016-4/wes-2016-4-RC2-supplement.pdf

**Supplement:**

[revised manuscript text omitted]

In this study ==we measure the wake position and the wind direction in the LES,== thus the unknown quantity is the magnitude of the wake deflection. For a fixed thrust coefficient, turbine site, wind speed and wind direction, the wake deflection is

15  assumed to be a function of the yaw angle $\gamma$ and the vertical veer and shear of the wind ==created by the atmospheric stability== $L$.

$$\Delta y_\gamma = \Delta y_\gamma(\gamma, L) \tag{2}$$

To find the relationship of $\Delta y_\gamma$ on the yaw angle and the atmospheric stability, the wake deflection is estimated from ==measurements== by estimating $\mu_y$ and $y_0$ for fixed $\gamma$ and $L$.

$$<\Delta y_\gamma>|_{\gamma,L} = <\mu_y(f_i, \Delta t)> - <y_0(\alpha_h(\Delta t))> \tag{3}$$

20  Here we consider ==that== $\mu_y$ depends on the ==method== $f_i$  the wake center position from the  and that both $\mu_y$ and $\alpha_h$ depend on the temporal averaging interval $\Delta t$. By doing so, we expect to have errors in the estimation of $\mu_y$ and $y_0$ that propagate into the error of $\Delta y_\gamma$. If the wake position is estimated following Eq.1 from averages of the measured hub height wind direction, we are expecting to make a statistical error related to the stochastic nature of the flow in the atmospheric boundary layer. We  this error by calculating the standard deviation of the mean wake deflection for different individual

25  estimations, each with a temporal averaging interval $\Delta t$.

$$\sigma_{<\Delta y_\gamma(\Delta t)>}|_{\gamma,L} = \left(\frac{1}{n-1}\sum_i^n (<\Delta y_\gamma(\Delta t)>_i - \overline{<\Delta y_\gamma(\Delta t)>})^2\right)^{1/2} \tag{4}$$

A qualitative error that can be made in the estimation of the wake deflection is a bias introduced by the method $f_i$ ==to== find the wake center position from the measured data. An evaluation of this bias is done by comparing different methods $f_i$.

[Figure]

**2.2 Estimating the wake displacement by the change of wind direction**

We consider the wind conditions at $x_1 = 2.5$ rotor diameter (D) upstream as reference inflow conditions to a wind turbine. This distance is chosen as it is consistent with the design standard of the IEC-61400-12-1 (2005) guidelines. More precisely our inflow information is hub height wind speed $u_h$ and wind direction $\alpha_h$ averaged at $x_1$ on a line of $\Delta y = 2\,$D perpendicular to the expected mean wind direction (Fig. 1). We choose cross stream averaged variables instead of a point measurement as we consider them more representative for the wind conditions for the whole wind turbine.

To estimate the wake displacement $y_0$ we extrapolate the wake along the wind direction. 
[revised manuscript text omitted]

---

## Author Comment (AC1)

**Estimating the wake deflection downstream of a wind turbine in different atmospheric stabilities: An LES study**

Lukas Vollmer, Gerald Steinfeld, Detlev Heinemann, Martin Kühn

August 9, 2016

In this document we answer to the comments of the reviewer RC1. We attached the new manuscript, as we found that this allows for a better traceability of our modifications. All changes on the manuscript are highlighted with a blue font color.

**Review**

The paper sets out to address by means of LES the effect of wind turbine yaw on the deflection of a turbine wake in three types of wind: neutral, stable and convective. The deliberate deflection of wakes as a means of reducing adverse affects on downwind turbines is of practical interest. The paper presents interesting results which deserve to be published once some improvements to the manuscript are made, as outlined below.

**Authors' comment:** *We thank the reviewer for the helpful comments and suggestions. We tried to incorporate all editorial modifications and language corrections and are very thankful for the detailed comments on these aspects. We hope we were able to answer all questions sufficiently in the text below.*

**Page 1**

Authors and affiliations - why superscript 1 in each?
**Authors' comment:** *Superscripts are now removed because all authors belong to the same institute.*

L9 Uncertainty in the
L11 increases with decreasing stability Does this mean increases with increasing stability (from unstable to stable)?
L17 can have a significant .. 10-15 rotor diameters or more downstream
L23 The latter
**Authors' comment:** *The text was edited according to the suggestions of the reviewer with exception of L11. The statement in the original text is correct as the variance of the estimation increases with decreasing atmospheric stability (from stable to unstable). See also Table 2 for verification.*

**Page 2**

L18 "even suggested considering" But, what is the sentence saying anyway?

**Authors' comment:** *Fleming et al. 2016 calculate the energy yield of wind farms with an engineering model and include a control of the yaw angle of the turbines to optimize the yield. They even go further and explore combinations of a new layout with application of yaw control to maximize the energy yield per area.*

| **Old Version** | **New Version** |
|---|---|
| Fleming et al. (2016) even suggest to consider the possibility of wind farm control in the construction of new wind farms. | Fleming et al. (2016) even suggest including power yield optimization by wind farm control in the design process of new wind farm layouts. |

L30 "influence of atmos stability on the control mechanisms themselves" This seems an odd way of putting it. - the issues is how the control would need to change with the change of wind conditions as influenced by stability. There is no influence on the control mechanisms as such.

**Authors' comment:**

| **Old Version** | **New Version** |
|---|---|
| Considering that the potential to improve wind farm efficiency through wind farm control appears to be dependent on atmospheric stability, little knowledge exists on the influence of atmospheric stability on the control mechanisms themselves. | Considering that the potential to improve wind farm efficiency through wind farm control appears to be dependent on atmospheric stability, little knowledge exists on how the control would need to adapt to changes of the wind conditions as influenced by atmospheric stability. |

This brief review refers to stability effects as observed from field measurements and computations. There has also been recent work via laboratory stratified flow wind tunnel experiments - e.g. in Boundary-Layer Met.: Chamorro, Hancock and co-workers. On meandering there has been work by Aubrun and co-workers in J Wind Engin and Ind Aerodyn. These could be mentioned too.

**Authors' comment:** *Following the suggestions of both reviewers we added a paragraph to the introduction which introduces recent work on the topic from wind tunnel experiments and numerical experiments.*

*" The observation of a change of wind farm performance with different atmospheric stability has been supported by wind tunnel experiments and numerical studies. It has been either related to a generally different level of turbulence (Hancock and Zhang, 2015) or to the presence of large scale fluctuations that enhance the so-called meandering of the wakes in less stable situations (Machefaux et al., 2015a; Larsen et al., 2015; Keck et al., 2014; Espana et al., 2011). Emeis (2010) and Abkar and Port-Agel (2013) argue that the thermal stratification above the wind farm becomes important for large wind farms as the vertical momentum transport becomes the only kinetic energy source to refill the wake deficit. Apart from the energy yield the structural loads on turbines in the wake also differ with atmospheric stability as they*

*are influenced by up-and downdrafts and large coherent structures in a CBL (Churchfield et al., 2012) and by sharp velocity gradients in an SBL (Bromm et al., 2016). "*

Figure 1 is confusing. With yaw there is only one change; the hub axis with respect to a reference mean wind direction - at say hub height. Fig 1a is sufficient, Fig 1b is not needed and adds confusion. Fig 1a is sufficient when there is wind veer. Also, the yaw as shown in the fig is negative. Perhaps this should be mentioned in the figure caption, or the figure redrawn with positive yaw illustrated. (Is there consistency in sign of yaw angle as used later in the paper?)

**Authors' comment:** *We agree with the opinion of the reviewer that the only change with yaw is the hub axis with respect to a reference mean wind direction. However, to introduce the uncertainty that arises from the estimation of the wind direction we have to include $y_0$ in the sketch. Based on the comment of the reviewer we chose to remove Fig. 1 (a) and to use only Fig. 1 (b), because all important variables are defined in this sketch. The yaw angle is consistently defined as positive for a clockwise turning of the rotor. We now mention that a neg. yaw angle is shown in the figure.*

**Page 3**

Section 2.1

This seems unnecessarily complicated. At any instant, t, there will be a velocity profile U(t, y) at given x. Over a short time internal $\Delta$t, there will be an short-period averaged profile U(t, $\Delta$t, y)), which will vary with t, except when $\Delta$t is very long in which case the profile will no longer be a function of t. U(t, $\Delta$t, y)), for given t, will vary with x, and so can be used to define the wake (for that t and $\Delta$ t). The long-term average ie large $\Delta$t will define a mean position varying with x. The short-term average will be a variation about this, but it does not make sense to refer to this as an error. The variation about the steady state is important is important as far as any downwind turbine is concerned.

**Authors' comment:** *Following the suggestions of the reviewer we simplified Section 2.1. We hope the new formulation is more intuitive to the reader.*

**Old Version**

To find the relationship of $\Delta y_\gamma$ on the yaw angle and the atmospheric stability, the wake deflection is estimated from measurements by estimating $\mu_y$ and $y_0$ for fixed $\gamma$ and $L$.

$$< \Delta y_\gamma > |_{\gamma,L} = < \mu_y(f_i, \Delta t) > - < y_0(\alpha_h(\Delta t)) > \tag{1}$$

Here we consider that $\mu_y$ depends on the method $f_i$ to find the wake center position from the measured data and that both $\mu_y$ and $\alpha_h$ depend on the temporal averaging interval $\Delta t$. By doing so, we expect to have errors in the estimation of $\mu_y$ and $y_0$ that propagate into the error of $\Delta y_\gamma$. If the wake position is estimated following Eq. 1 from averages of the measured hub height wind direction, we are expecting to make a statistical error related to the stochastic nature of the flow in the atmospheric boundary layer. We calculate this error by calculating the standard deviation of the mean wake deflection for different individual estimations, each with a temporal averaging interval $\Delta t$.

$$\sigma_{<\Delta y_\gamma(\Delta t)>}|_{\gamma,L} = \left( \frac{1}{n-1} \sum_i^n (< \Delta y_\gamma(\Delta t) >_i - < \Delta y_\gamma(\Delta t) >)^2 \right)^{1/2} \tag{2}$$

A qualitative error that can be made in the estimation of the wake deflection is a bias introduced by the method $f_i$ to find the wake center position from the measured data. An evaluation of this bias is done by comparing different methods $f_i$.

**New Version**

To find the relationship of $\Delta y_\gamma$ on the yaw angle and the atmospheric stability, the mean wake deflection is estimated from measurements by estimating $\mu_y$ and $y_0$ for fixed $\gamma$ and $L$.

$$< \Delta y_\gamma > |_{\gamma,L} = < \mu_y(f_i) > - < y_0(\alpha_h) > \tag{3}$$

Here we consider that $\mu_y$ depends on the method $f_i$ to find the wake center position from the measured data. To calculate the temporal variation of the wake deflection we divide the time series into shorter intervals $\Delta t$ and calculate the variance of this individual estimates about the mean.

**Page 4**

Section 2.3

It seems slightly odd to use wake profile forms that are "symmetrical" when the results, as might be anticipated (and are) clearly not symmetrical. Could other measures be used? Short period averaged-centre as defined by a velocity minimum, and wake half width?

**Authors' comment:**   *The three different measures to define the wake center, which are used in this manuscript are chosen because the authors believe they are reproducible by full scale measurements with either lidar systems or with the power measurements of a row of two wind turbines. Our opinion is that a velocity field that can be fitted to a Gaussian-like shape*

*is easier to measure than finding the absolute velocity minimum in the wake.*
*The averaged profile at hub height is quite symmetrical for all cases (see Fig 5,9,12 d), thus we expect no difference by using measures as a velocity minimum or the wake half width. For the y-z cross sections through the wake, a wake half width is difficult to define but choosing the velocity minimum as wake center would definitely lead to different results in this case (see Fig 5,9 a-c).*

**Page 6**

L 11 Why mention of humidity? This is an added complication if this is being modelled too.
**Authors' comment:** *The model is able to include humidity but that was not done in the present study. The text is changed to prevent distraction of the reader.*

| **Old Version** | **New Version** |
|---|---|
| The statistics of the steady turbulence that develops after some spin-up time depend on the initial boundary conditions provided for the fluid, e.g. temperature and humidity, and the boundary conditions during the simulation, e.g. surface heat fluxes. | The statistics of the steady turbulence that develops after some spin-up time depend on the initial conditions provided for the fluid, e.g. the temperature profile, and the boundary conditions during the simulation, e.g. surface heat fluxes. |

Section 2.7
Mesh size of 5m seems very large.
**Authors' comment:** *We used a comparable mesh size to Witha et al. (2014), Dörenkämper et al. (2015) and Mirocha et al. (2015). In Wu and Porté-Agel (2015) "Modeling turbine wake and power losses within a wind farm using LES: An application to the Horns Rev offshore wind farm", which also use the ADM-R model, a mesh size of about 16 m is used. All simulations were run with the same mesh resolution for comparability. For the simulation of the stable boundary layer a smaller mesh size would lead to slightly different turbulent fluxes but here we followed the conclusion by Beare et al. (2004) that a mesh size of about 6 m is able to resolve the majority of the fluxes.*

Surely, this should say, $\Delta x$, $\Delta y$, $\Delta z$, rather than just $\Delta x$?
**Authors' comment:** *The text now says "$\Delta = 5\,m$"*

Section 2.8
L31. Shouldnt the points made here be made in section 2.7?
**Authors' comment:** *In contrast to Section 2.7, Section 2.8 deals with the setup of the main simulations with presence of turbines. As the setup differs from the precursor simulations described in Section 2.7 we feel it is necessary to devide these two topics into two separate subsections.*
*We attribute the reviewer's comment partially to the points we make about the necessary size of the domain of the CBL. Section 2.7 contains a comment about the different domain sizes, we removed the reason for the different size in Section 2.8 because they do not necessarily belong to the description of the main simulations.*

| **Old Version** | **New Version** |
|---|---|
| In the CBL the necessary domain width $L_y$ to consider the energy contained in the large eddies is more than 6 times larger than the minimum size of $L_y^{min}$. | In the CBL the domain width $L_y$ is more than 6 times larger than the minimum size of $L_y^{min}$. |

**Page 7**

Fig 3. It would be very useful to have the temperature profiles in this figure. This should be included as it is basic to non-neutral flow. (Suggest use I rather than TI; I is commonly used in meteorology.) It would also be helpful to include the cross-flow as an angle.

**Authors' comment:** *Temperature profiles were added to Fig 3, the profile of the v-component was replaced by a profile of the cross-flow as an angle. In the cited literature I ( e.g. Wharton et al. 2012 ) is used as well as TI ( e.g. Dörenkämper et al. 2014, Hansen et al. 2012 ) for turbulence intensity. Because of the expected audience of the journal we decided to use the term TI, which might be more intuitive for non-meteorologists.*

**Page 8**

L17 does this comment not apply to CBL as well as SBL and NBL?
**Authors' comment:**

| **Old Version** | **New Version** |
|---|---|
| Note that due to the cyclic lateral boundary conditions in all simulations, the turbines in the SBL and NBL are in principle also part of an infinite row along $y$. | Note that due to the cyclic lateral boundary conditions, the turbines in all simulations are part of an infinite row along $y$ separated by more than $L_y^{min}$. |

L20 of an upstream observer looking downstream. is helpful to be quite clear.
**Authors' comment:** *Edited according to the suggestion of the reviewer*

**Page 9**

Fig 5. Presumably, the differences between a) and c) not being anti-symmetric is rotation in the wake. Which way is the turbine rotating, and therefore the wake in the opposite direction? Does this assist or oppose wind veer? I dont think there is mention of this.
Figs 5, 9 and 12. Say or show which way the rotor is rotating.
**Authors' comment:** *The sense of rotation of the rotor is mentioned in Section 3.1. For clarity we also included a remark in Section 2.6 in the description of the turbine model. In general the rotation of the rotor is opposing wind veer. The reason for the asymmetry is also related to the induction by yaw either opposing the rotation in the upper or in the lower rotor half. We changed the sign of the cross-stream component in Fig. 7 , so that the sense of the rotation of the wake is more intuitive. We think this figure gives a rather good impression about the rotation of the wake. We decided to not show or say in every figure which way the rotor is rotating, because we found it rather distracting for the reader as the wake is rotating*

*with the opposite sense of rotation. Following sentence was added in Section 3.1: "Figure 7 (a,c) show that this cross stream momentum is either opposing the rotor rotation below or above the hub, which, together with the influence of wind veer, leads to the asymmetries further downstream as evident in Fig. 5 (a,c)."*

Fig 6. Why not show these as lines rather than points - perhaps using solid, dashed and broken lines?
**Authors' comment:** *Plots were changed*

L17. The resulting votices.. How does the vorticity get there, or is it an irrotational rotating motion?
**Authors' comment:** *We removed the rather misleading term "vortices"*

| Old Version | New Version |
|---|---|
| The resulting vortices are responsible for the varying magnitude of lateral displacement at different heights and the crescent shape of the wake further downstream. | The opposing cross stream velocities appear to be responsible for the varying magnitude of lateral displacement at different heights and the crescent shape of the wake further downstream. |

L26 The fitting method used..
L32 .. the extent..
**Authors' comment:** *Edited according to the suggestion of the reviewer*

**Page 11**

L5 Paragraph. Isnt it just that the turbulence intensity is larger in the CBL case? Are these large scale structures not seen in the NBL and SBL to any degree?
**Authors' comment:** *The large scale structures as evident in fig. 3(b) and fig. 4 are characteristic for a moderate convective boundary layer. These features are described in Section 2.7, we added an additional comment at this point*

**Page 12**

L10 paragraph. This seems to overlook the much large v at low frequency, as shown in figure 4.
**Authors' comment:**

| Old Version | New Version |
|---|---|
| The reason for the inability to use our method to find the wake deflection is most probably a deviation of the ambient flow downstream of the turbine from the flow upstream of the turbine, that can not be described by the taylor theorem. | Apparently, the stochastic fluctuation of the wake caused by the large fluctuations of the cross stream component are superimposing the trajectory change of the wake caused by the induction of the turbine to a degree that the latter signal is not detectable. |

L25 .. measurement device used.
**Authors' comment:** *Changed*

**Page 16**

Table 3. Parameters are not introduced, defined and discussed in the text.
**Authors' comment:** *We relate to the parametric description of the wake deflection as derived in Jimenez (2010) and Gebraad et al.(2014). The shown parameters are defined in Gebraad et al. and are only briefly discussed in our manuscript. Nevertheless we feel that they can be a valuable contribution to the attempt to parameterize the wake trajectory in different stability regimes. We added a sentence on the consequence of different wake deflection parameterizations for different atmospheric stability:*
*"Gebraad et al. (2016) show that the energy yield of a small wind farm can be well predicted by a simplified parametric model, which is fitted to simulated atmospheric conditions of neutral stability, and that the energy yield of a small wind farm can be improved by more than 10 percent for certain scenarios. Assuming the same parameters for the stable wind field from our study would lead to a miscalculation of the wake position which corresponds to a yaw induced deflection by a yaw angle of about 10°. Thus, the described parametrization of the model would likely propose an unfavorable control for stable situations."*

**Page 22**

Fig 14. y/D and x/D, not as shown
**Authors' comment:** *y[D] and x[D] was used throughout the whole manuscript. We changed it to x/D and y/D*

[revised manuscript text omitted]

---

## Author Comment (AC2)

**Estimating the wake deflection downstream of a wind turbine in different atmospheric stabilities: An LES study**

Lukas Vollmer, Gerald Steinfeld, Detlev Heinemann, Martin Kühn

August 9, 2016

In this document we answer to the comments of the reviewer number 2. We attached the new manuscript, as we find that this allows for a better traceability of our modifications. All changes on the manuscript are highlighted with a blue font color.

**Review RC2**

Intentional yaw misalignment of wind turbines as a way to control/optimize wind farm performance is indeed a relevant scientific topic. One part of this question is the potential for increased power production  another aspect is the increased loading following from WT yaw operation, which in principle is a cost. This paper links to the first mentioned aspect only and discusses whether or not the required (precise) knowledge of downstream wake trajectories can be achieved. This is crucial for the relevance of wind farm yaw control. The paper ends by concluding that for highly convective flow conditions, wind farm yaw control cannot be recommended, whereas the approach might be feasible under SBL conditions. However, there is no firm conclusion on the perspectives for the yaw-control approach under NBL conditions - i.e. whether sufficient precise knowledge of downstream wake trajectories is likely achievable in practice. The scientific approach is sound, but the representativeness of the selected SBL/CBL cases should be discussed. References to relevant related work are in general good, but a few additional references should be added. The suggested references are found in the attached detailed pdf-review at the relevant passages in the manuscript. The detailed review moreover indicates a few misprints, asks for a few additional discussions/clarifications and suggests some editorial modifications.

**Authors' comment:** *We thank the reviewer for the very helpful comments and suggestions. We tried to incorporate all editorial modifications and language corrections and are deeply grateful for the detailed comments. We identified as one main criticism the sole focus of this study on the potential for increased power production while neglecting the increased loading following from wind turbine yaw operation. In addition we should comment on a conclusion on the perspectives for the yaw-control approach and how representative the meteorological conditions of the simulations are.*
*The cost of increased loading is mentioned in the new manuscript, but we also tried to be more precise about the findings of this study which we consider independent from the question whether energy yield or loading should be optimized in a wind farm. We think our main*

*contribution is on the consideration of uncertainties about the inflow wind conditions and its effect on the wake trajectory that has to be considered, if any kind of wind farm control is to be further developed towards industrial application. A profound cost/ benefit analysis of the wind farm control by yaw misalignment is not in the scope of this study. The introduction, discussion and conclusion sections of the manuscript were edited to highlight these aspects more. Gebraad et al. (2016) show that the energy yield of a small wind farm can be well predicted by a simplified parametric model, which is well tuned to the atmospheric conditions (neutral stratification) and that the energy yield can be improved by more than 10 percent for certain scenarios. Thus, we are optimistic that a sufficient precise knowledge can be achieved with adaptive models and the adequate supporting measurement system. The selected SBL/CBL cases are of course only examples but do not represent extraordinary conditions. Several studies show that non-neutral wind conditions are usually occurring at wind farms, e.g. Wharton and Lundquist (2012) consider only about 20 % of the measured conditions at an onshore site as neutral, while Hansen et al. (2012) classify about 20 - 30 % as neutral at an offshore wind farm. In the following the individual comments on the manuscript are answered.*

**Page 1**

Number 1: the inflow
Number 2: a source of
Number 3: Control
Number 4: the ambient
**Authors' comment:** *Edited according to the suggestion of the reviewer*

**Page 2**

Number: 1
Yes, but I guess that neutral conditions still prevails at high wind speeds, where the wind farm produces the most.
**Authors' comment:** *Indeed neutral or near neutral conditions prevail at hub height wind speeds larger than about 12 m/s where the wind farm produces most (see e.g. Barthelmie et al. 1999, Wharton and Lundquist 2012, Hansen et al. 2012). There are several reasons which lead us to believe that a measure of atmospheric stability or wind shear and veer influenced by atmospheric stability is important for wind farm control. First, at about 12 m/s the thrust coefficient decreases for most turbine types and the power curve is much flatter than for lower wind speeds. Thus the losses due to wakes become smaller when you surpass the rated wind speed of the upwind turbine. Second, the structural loads increase with increasing wind speeds and at high wind speeds it becomes unlikely that the turbine manufacturers are willing to further increase the loads by yawing the turbine. Third, there are only few sites for which the most probable wind speed is above 12 m/s. Most studied onshore sites in literature have the most occurring wind speeds in the range between 6-10 m/s where thermal fluxes are still important.*

Number: 2
There is a third possibility - dynamically change turbine power and thrust set-points with time scales in the range of 20 to 30 seconds, supposedly leading to faster wake recovery: Goit J, Meyers J. Optimal control of energy extraction in wind-farm boundary layers. Journal of Fluid Mechanics 768:5-50, 2015.

**Authors' comment:** *The approach made in the paper of Goit and Meyer is certainly interesting as it reveals which processes influence the wake recovery in a large wind farm, but we can't yet see how this control can be implemented in a full size wind farm. In our study we want to focus on solutions for wind farm control that are close to be implemented in real wind farms. That's why we decided not to cite the contribution by Goit and Meyer.*

Number: 3
+ an increase in loading for the yaw control approach!
**Authors' comment:** *Our goal was not to conclude on the costs and benefits of wind farm control but the cost of increased loading is obviously not negligible. We added a short comment on this topic.*

**Page 3**

Number 1: also to
**Authors' comment:** *Changed*

Number 2: Reformulate
**Authors' comment:**

| **Old Version** | **New Version** |
|---|---|
| In this study we measure the wake position and the wind direction in the LES, thus the unknown quantity is the magnitude of the wake deflection. | The advantage of LES is that the wake position and the wind direction can be assessed directly from the flow field to estimate the unknown deflection of the wake by the yawed turbine. |

Number 3: Veer and shear are not created, but affected, by ABL stability. Why not just state that $\Delta y$ depends on yaw angle and ABL conditions as stated in your eq. (2) Does "L" refer to Monin-Obukhov length?
**Authors' comment:**

| **Old Version** | **New Version** |
|---|---|
| For a fixed thrust coefficient, turbine site, wind speed and wind direction, the wake deflection is assumed to be a function of the yaw angle $\gamma$ and the vertical veer and shear of the wind created by the atmospheric stability $L$. | For a fixed thrust coefficient, turbine site, wind speed and wind direction, the wake deflection is assumed to be a function of the yaw angle $\gamma$ and the atmospheric stability, e.g. given by the Monin-Obhukov length $L$. |

Number 4: simulated flow fields(?)
**Authors' comment:**

| **Old Version** | **New Version** |
|---|---|
| To find the relationship of $\Delta y_\gamma$ on the yaw angle and the atmospheric stability, the wake deflection is estimated from measurements by estimating $\mu_y$ and $y_0$ for fixed $\gamma$ and $L$. | The relationship of $\Delta y_\gamma$ on the yaw angle and the atmospheric stability is estimated from multiple LES with different $\gamma$ and $L$. |

Number 5 - 13:
**Authors' comment:** *The whole part of the text was changed following the suggestions of both reviewers*

**Page 4**

Number 1: Please motivate this choice (induction, ...)
**Authors' comment:**

| **Old Version** | **New Version** |
|---|---|
| This distance is chosen as it is consistent with the design standard of the IEC-61400-12-1 (2005) guidelines. | This distance is chosen as the wind field closer to the turbine might be modified by the induction of the rotor (IEC-61400-12-1, 2005). |

Number 2: extending
**Authors' comment:** *Changed*

Number 3: -
Number 4: turbine rotor
Number 5: -
**Authors' comment:** *3-5 changed*

Number 6: assume a wake advection trajectory (?)
**Authors' comment:**

| **Old Version** | **New Version** |
|---|---|
| To estimate the wake displacement $y_0$ we extrapolate the wake along the wind direction. | To estimate the wake displacement $y_0$ we assume an advection of the wake with the ambient wind. |

Number 7: $\Delta x_2$
**Authors' comment:** *We consider $\Delta x_2$ as the distance between the rotor surface and a surface at $x_2$ (see Fig. 1). Thus we think the current notation to use $x_2$ is correct.*

Number 8: or: approach
**Authors' comment:** *Changed*
Number 9: Gaussian-like ... Gaussian functions integrate to 1 identically
**Authors' comment:** *Changed*
Number 10: a measure of the width of the wake
**Authors' comment:** *Changed*
Number 11: circular
**Authors' comment:** *Changed*
Number 12: whereas for
**Authors' comment:** *Changed*

**Page 5**

Number 1: Why not include info on the vertical deficit location in analogy with approach 2)
described in eq. (6) - i.e. allow center of deficit to be located at other heights that $z_h$?
**Authors' comment:** *The approach of available power is made, because it should be representative for the energy yield of a downstream turbine. These are usually at same height. The info about the vertical location from approach 2) was not further used in the study.*

Number 2: of respectively
**Authors' comment:** *Changed*

Number 3: identify
**Authors' comment:** *Changed*

**Page 6**

Number 1: Describe and motivate the smearing function used.
**Authors' comment:** *The smearing function is introduced in detail in Doerenkaemper et al. (2015) and Bromm (2016). We added a remark on the choice of the value of the regularization parameter, which follows a study by Troldborg et al. (2014). In internal sensitivity studies we found that a value of twice the grid size is a good choice for the regularization parameter as also concluded by Troldborg. This value represents a good compromise between local resolution of the blade properties and smoothness of the force distribution.*

| Old Version | New Version |
|---|---|
| The forces are scaled for a three bladed turbine and are afterwards projected onto the grid of the LES by a smearing function. | The forces are scaled for a three bladed turbine and are afterwards projected onto the grid of the LES by a smearing function with a Gaussian kernel as described in Dörenkämper (2015b). In internal sensitivity studies we found that a value of twice the grid size is a good choice for the regularization parameter as also concluded by Troldborg et al. (2014). |

Number 2: but with

**Authors' comment:** *Changed*

**Page 7**

Number 1: Are the resulting SBL and CBL fields representative? Could you quantify the resulting stability conditions in terms of simple measures as e.g. Monin-Obuhkov length or Richardson number?

**Authors' comment:** *We added some remarks on the representativeness of the simulated boundary layers and the quantification of the stability conditions which are mentioned in Tbl. 1.*

*" Shear coefficient $\alpha_s = 0.30$ and Monin-Obhukov length $L = 170\,m$ correspond to a stable to highly stable stability class following Wharton and Lundquist (2012).*
*...*
*The CBL represents a rather moderate convective boundary layer with $L = -180\,m$ and a ratio between the boundary layer height $z_i$ and $L$ of $z_i/L = -3.6$.*
*...*
*The meteorological conditions of the CBL and SBL simulation cases are regularly occurring at wind farm sites ( Hansen et al., 2012, Vanderwende and Lundquist ,2012; Wharton and Lundquist, 2012). Numerical simulations comparable to the CBL and NBL case are studied in Churchfield et al. (2012), while Mirocha et al. (2015) simulate even stronger stable and convective conditions, which are motivated by measured events. "*

**Page 8**

Number 1: $\Delta x_2$ (?) ... cf. fig. 1
Number 2: $\Delta x_2$ (?)
**Authors' comment:** *See earlier comment on this topic*

Number 3: take the smallest values
**Authors' comment:** *Changed*

**Page 9**

Number 1: Gaussian-like
Number 2: is likely to
**Authors' comment:** *Both changed*

Number 3: $\Delta x_2$ (?)
**Authors' comment:** *See earlier comment on this topic*

**Page 10**

Number 1: Which of the two suggested mechanisms that is the most important could maybe be investigated by reversing the rotor rotation direction.

**Authors' comment:** *We additionally conducted simulations with a reversed rotor rotation. These were now added in Fig. 10 to show the influence of the rotor rotation. The text was also modified as follows:*

**Old Version**

As apparent in Fig. 9, the wake center is located a little higher than hub height, therefore the ambient wind direction at wake center height is slightly towards the right. Both effects would explain the difference between the wake deflection in the SBL and the NBL.

**New Version**

Trajectories of simulations with a reversed rotation of the rotor show that the sense of rotation is not exclusively responsible for the bias to the right as this would lead to a mirroring of the trajectories about the wind direction for opposite rotor rotations (Fig. 10). As apparent in Fig. 9, the wake center is located a little higher than hub height, therefore the ambient wind direction at wake center height could also lead to a slight advection towards the right. Thus both effects seem to be responsible for the difference between the wake deflection in the SBL and the NBL.

Number 2: are
**Authors' comment:** *Changed*

Number 3: Have been "validated" using full-scale experimental data in: G.C. Larsen et al. (2015). Wake meandering under non-neutral atmospheric stability conditions theory and facts. Journal of Physics: Conference Series (Online), Vol. 625, 012036. and in Machefaux, E. et al. (2015). An experimental and numerical study of the atmospheric stability impact on wind turbine wakes. Wind Energy, DOI 10.1002/we.1950.
**Authors' comment:** *We added citations of the work of Larsen et al. and Machefaux et al. regarding the theory of the meandering frame of reference.*

**Page 11**

Number 1: or: identified
**Authors' comment:** *Changed*

Number 2: seems (?)
**Authors' comment:** *Changed to "appears"*

Number 3: reformulate
**Authors' comment:**

**Old Version**

The consideration of the advection time $\tau$ increases the similarity of the wind conditions most in the CBL, where especially the variance of the wind direction is large.

**New Version**

A shift of the downstream timeseries by $\tau$ has the largest effect on the similarity of the wind conditions in the CBL, where especially the variance of the wind direction is large.

Number 4: A
Number 5: or: aiming
Number 6: wake
Number 7: or: for
**Authors' comment:** *4-7 Changed*

**Page 12**

Number 1: correlation
**Authors' comment:** *Changed*

Number 2: or: impossible
**Authors' comment:** *Yaw control to deflect the wake can still be applied but it apparently does not make much sense*

Number 3: ... but why should the Taylor assumption be more critical for CBL than for SBL/NBL? - please explain
**Authors' comment:** *With the approaches we make in the manuscript we can not fully explain why the Taylor assumption does not work as well in the CBL as in the other simulations. Based on the findings in Fig. 15 and the snapshots of the flow as shown in Fig. 14 we think it is mainly related to the averaging over larger cross stream structures and the advection of these structures into the domain downstream of the turbine that are not covered by the measurement upstream of the rotor. We changed the text slightly at this point to highlight the presence of these large scale structures.*

| **Old Version** | **New Version** |
|---|---|
| The reason for the inability to use our method to find the wake deflection is most probably a deviation of the ambient flow downstream of the turbine from the flow upstream of the turbine, that can not be described by the taylor theorem. | Apparently, the stochastic fluctuation of the wake caused by the large fluctuations of the cross stream component are superimposing the trajectory change of the wake caused by the induction of the turbine to a degree that the latter signal is not detectable any more. |

Number 5: Disregarding ABL stability issues, advection of wake deficits is discussed in: Machefaux et al. (Empirical modeling of single-wake advection and expansion using full-scale pulsed lidar-based measurements; Wind Energ. 2015; 18:20852103)
**Authors' comment:** *We included a remark on the current work by Machefaux et al. on this topic.*

Number 6: Yes - why not use the average over an imaginary upstream rotor ... which could also in practice be supported by lidar systems.
**Authors' comment:** *This is certainly an interesting option for the further research on this topic. It would be interesting to see how precise temporal averages can be achieved with this method on relevant time scales.*

**Page 13**

Number 1: LES simulations
**Authors' comment:** *As LES already includes the term simulations we used the term "case studies" instead.*

Number 2: Given the fact that nacelle based lidar systems should be applied along with some sensor (sonic?) that enables quantification of ABL stability conditions - to distinguish CBL from SBL/NBL - do yaw based WF control then in your opinion have a future in full-scale WF's??
**Authors' comment:** *We think it is difficult to judge on the application of wind farm control without doing a profound cost/benefit evaluation. We emphasize that a control without good measurements is not possible, a lidar is not necessarily part of this system. A different approach could be to optimize wind farm performance with consideration of the uncertainties coming from the measurement system and the stochastic nature of the flow.*

Number 3: or: correlation
**Authors' comment:** *Changed*

Number 4: wake position
**Authors' comment:** *Changed*

Number 5: It should be mentioned that even in ideal cases, where intentional wake deflection can be controlled, the potential improvement in wind farm power production comes with a price, as the loading (and thus structural degradation) of the yawed turbine(s) is increased.
**Authors' comment:** *A comment on the increased loading was added.*

[revised manuscript text omitted]